# Evaluation of global outbreak surveillance performance for high pathogenicity avian influenza and African swine fever

Younjung Kim [1,2] ✉, Guillaume Fournié [3,4,5], Paolo Tizzani [6], Gregorio Torres [6], Raphaëlle Métras [1], Dirk Pfeiffer [3,7] & Pierre Nouvellet [2]

Timely outbreak notification is critical for successful disease control. For high pathogenicity avian influenza (HPAI) and African swine fever (ASF), surveillance performance within and across countries remains largely unknown, despite their continued global spread. We assessed surveillance performance in reporting HPAI outbreaks (2020–2023) and ASF outbreaks (2016–2023) amongst World Organisation for Animal Health (WOAH) member countries/ territories using WOAH outbreak notification data. We employed a modelling approach using the number of fatalities reported at initial notification as a performance measure, where fewer fatalities relative to the size of the affected premise—after accounting for other premise-level and country-level factors— were assumed to indicate better performance. For both diseases, fatalities were strongly associated with factors including country/territory, number of susceptible animals, premise type (commercial farms, backyard farms, and villages), season, and spatiotemporal outbreak clustering. The number of susceptible animals and country/territory explained the most variance in fatalities. While a few countries/territories appeared to perform exceptionally well or poorly, significant overdispersion suggested substantial heterogeneities within countries/territories after controlling for other factors. Our findings highlight the need for more targeted national and global efforts to strengthen animal health surveillance capacities.

High pathogenicity avian influenza (HPAI) and African swine fever (ASF) viruses are major concerns for animal health, with HPAI viruses also posing a continuous threat to human health. Both viruses cause up to 100% mortality in infected populations and can spread across borders through wildlife movement and the trade of animals and animal products[1,2].

In recent years, the transboundary spread of HPAI and ASF viruses has become particularly prominent. The newly emergent wild bird-adapted HPAI H5N1 clade 2.3.4.4b viruses have quickly become predominant worldwide since their first identification in Europe in 2020[3]. In 2022 alone, H5N1 viruses were associated with the reported death of 141 million domestic and wild birds across 85 countries and territories[4]. ASF virus has spread from sub-Saharan Africa to Europe, Asia, and the Americas, with continued westward expansion in Europe and widespread spread in Asia in the past decade. From January 1 2022 to June 24 2024, the virus was identified in 57 countries across Africa,

[1]Sorbonne Université, INSERM, Institut Pierre Louis d'Épidémiologie et de Santé Publique (IPLESP), UMRS Paris, France. [2]Department of Ecology and Evolution, School of Life Sciences, University of Sussex, Brighton and Hove, UK. [3]Department of Pathobiology and Population Sciences, Royal Veterinary College, London, UK. [4]Université de Lyon, INRAE, VetAgro Sup, UMR EPIA Marcy l'Etoile, France. [5]Université Clermont Auvergne, INRAE, VetAgro Sup, UMR EPIA Saint-Gènes-Champanelle, France. [6]World Organisation for Animal Health, Paris, France. [7]Centre for Applied One Health Research and Policy Advice (OHRP), City University of Hong Kong, Kowloon, Hong Kong SAR, China. ✉e-mail: younjung.kim@stats.ox.ac.uk

Americas, Asia, and Europe, resulting in the reported loss of more than 1,406,000 animals[5].

These global trends highlight an urgent need for international, regional, and national responses to curb the transmission of these viruses[6,7]. An important consideration when coordinating such responses would be to understand the variability in the performance of animal health surveillance systems across countries/territories. This variability may stem from differences in the capacity of veterinary services along with other factors, including geographical locations, social and cultural practices, economic conditions, and political environments. Consequently, animal health surveillance performance is likely to be influenced not only by the specific pathogen and type of animal production concerned but also by the unique circumstances of each country/territory or related entity. Therefore, gaining a better understanding of the extent of this variability and associated factors should help identify priority areas to improve the effectiveness and efficiency of surveillance strategies against the transboundary spread of pathogens. However, this task is challenging, mainly due to the paucity of standardised data that allows for the comparison of animal health surveillance performance between regions managed by independent veterinary services[8].

In this study, we aimed to characterise the variability in HPAI and ASF outbreak surveillance performance within and between the member countries/territories of the World Organisation for Animal Health (WOAH). In most HPAI and ASF outbreaks, due to their highly fatal nature, an atypical mortality pattern often first raises alarms and leads to pathogen detection, rather than clinical signs that can be easily overlooked or misdiagnosed. Considering these epidemiological features, our Bayesian modelling framework assessed outbreak notification performance for premises with domestic animals (villages, backyard farms, and commercial farms), based on the extent of fatalities reported at initial outbreak notification to the WOAH through its World Animal Health Information System (WAHIS)[9]. The framework incorporated premise size and other epidemiological variables at the premise and country levels, under the assumption that, after controlling for those variables, a lower number of fatalities relative to premise size reflects better surveillance performance in outbreak notification. We focused on two specific periods: from 2020 to 2023 for HPAI outbreaks, corresponding to the global spread of HPAI H5N1 clade 2.3.4.4b viruses[4], and from 2016 to 2023 for ASF outbreaks, corresponding to the spread of the ASF virus in Eastern Europe and Russia, followed by its introduction into Asia and the Americas[5].

Our modelling framework was built upon the following three parameters, termed zero-fatality probability, fatality slope, and fatality threshold. The definition, underlying assumptions, and implications of these fatality metrics are summarised in Table 1 and depicted in Fig. 1 (see the "Methods" for details). Importantly, our modelling framework hypothesised that the epidemiological context of outbreaks can affect all of these fatality metrics. Therefore, each of these metrics was modelled as a function of the number of susceptible animals and other epidemiological variables considered to impact surveillance and pathogen transmission, including country/territory, premise type, season[10,11], cluster occurrence of outbreaks, income status, and quantitative veterinary service capacity, with the last three measured at the country/territory level. We estimated the fatality metrics and the effect of these contextual variables on the fatality metrics by fitting our model to data on the number of dead animals recorded at the initial notification of HPAI and ASF outbreaks to the WOAH WAHIS (hereafter referred to as fatality data) within a Hamiltonian Monte Carlo (HMC) Bayesian framework. Finally, based on the best-fitting model estimates, we assessed the outbreak notification performance of each country/territory by comparing its expected number of deceased animals at initial notification with the number expected assuming the average performance across all analysed countries/territories.

## Results

The analysis included 6,266 HPAI outbreaks in birds (with "birds" as the host in the WOAH WHAIS) reported between January 2020 and December 2023 from 35 countries/territories, and 10,461 ASF outbreaks in pigs (with "swine" as the host) reported between January 2016 and December 2023 from 26 countries/territories (Figs. S1, S2). These outbreaks were notified from villages, backyard farms, or commercial farms. The analysis included a median of 79 HPAI outbreaks (interquartile range [IQR]: 20–189.5) and 89 ASF outbreaks (IQR: 31–304.8) per country/territory. Country/territory names were anonymised in accordance with WOAH guidance. Across the countries/territories, 54.7% of the HPAI outbreaks and 52.0% of the ASF outbreaks were classified as occurring within statistically significant spatiotemporal clusters. The countries/territories had a median of 4 clusters (IQR: 1–4.75), with a median cluster duration of 2 months (IQR: 1–3.5 months). The information on bird and swine species was not available in the WOAH WAHIS. The outbreaks with no information on the number of susceptible animals, the number of dead animals, or both, were excluded (see Tables S1, S2 for the proportion of these outbreaks by country/territory).

### Variability in fatalities at initial outbreak notification

For HPAI, after accounting for premise size (the number of susceptible animals), the fatality data were best explained when each of the fatality metrics was differentiated by the following variables: (a) country/territory, (b) premise type (commercial farm or backyard farm/village), with or without (c) outbreak cluster by income status (whether an outbreak was in space-time clusters of outbreaks in a given country/territory, with distinct parameters applied based on the country/territory's income level; see the "Methods" for details) and (d) season (winter, spring, summer, or autumn) (Table 2). The model comparison using the leave-one-out information criterion (LOOIC) indicated no statistically significant differences between these models (Table 2). We selected the full model, which included all the above variables (a to d), as the best-fitting model, given that several categories within these variables were significantly associated with the fatality metrics (Table 3).

The HPAI fatality data exhibited a high level of overdispersion, with a median estimate of the overdispersion parameter, $k$, being 0.57 (95% highest density interval [HDI]: 0.55 to 0.59). The best-fitting model explained 39.1% of the deviance in the fatality data, a substantial increase from 15.2% explained by the baseline model, which assumed that the number of fatalities at initial notification was influenced only by premise size. Beyond premise size, most of the explained variability was attributed to country/territory, with the deviance explained increasing from 15.2% to 37.7% when country/territory was added to the baseline model. When other variables were sequentially added, the deviance explained increased only marginally (a 0.6% difference by premise type, 0.5% by outbreak cluster by income status, and 0.3% by season).

For ASF, the full model including (a) country/territory, (b) premise type (commercial farm, backyard farm, or village), (c) outbreak cluster by income status, and (d) season explained fatality data significantly better than other models, after accounting for premise size (Table 4).

The ASF outbreaks exhibited overdispersion in the number of fatalities at initial notification (median: 1.78, 95%HDI: 1.71–1.85), but to a lesser extent than for HPAI. The best-fitting model explained 57.7% of the deviance (Table 4), compared to 35.6% explained by the baseline model, which assumed that fatalities at initial notification only depended on premise size. Country/territory contributed the most to explaining the deviance, with 54.0% explained when it was added to the baseline model. The addition of premise type led to a 3.7% increase, but the addition of season and outbreak cluster by income status did not exhibit noticeable changes.

**Table 1 | Definitions and implications of fatality metrics**

| Fatality metrics | Definition | Assumption | Implication |
|---|---|---|---|
| Zero-fatality probability (Fig. 1a,c) | The probability of notifying outbreaks with zero fatalities. | In early outbreak stages, a pathogen can be detected before unusual mortality patterns arise, for example, through testing animals following nearby outbreaks or upon observing clinical signs[8,35]. In such cases, outbreaks would be notified without fatalities, or fatalities may not be differentiated from background mortality. | The zero-fatality probability provides insights into performance in early warning and detection. |
| Fatality slope (dashed lines in Fig. 1b, d) | The rate at which the number of fatalities at notification increases with premise size before plateauing. | In more advanced outbreaks where deaths are reported, larger susceptible host populations are likely to experience more fatalities due to the sheer size of the exposed population. This assumes that within-farm transmission is primarily density-dependent—larger premises are likely to house more animals per group, leading to transmission initially occurring within groups before potentially spreading further[16]. Additionally, small proportions of dead animals in such large populations could go unnoticed, thereby likely leading to more fatalities when outbreaks are recognised. | It reflects the effectiveness of biosecurity measures and surveillance in detecting outbreaks that have progressed enough to cause fatalities. For example, a steeper slope indicates that fatalities increase rapidly with premise size. This suggests that either biosecurity measures are weak, allowing fast disease transmission, or surveillance is ineffective, detecting outbreaks only after significant fatalities have occurred. Given the role of surveillance, the fatality slope can also serve as an indicator of early warning and detection performance, with a lower slope suggesting that outbreaks are detected earlier with fewer fatalities. |
| Fatality threshold (dotted lines in Fig. 1b, d) | The maximum number of fatalities at notification (i.e., the plateau). | As the number of susceptible animals (i.e., premise size) grows further, the correlation between premise size and the number of fatalities may weaken and could disappear. Thus, beyond a certain premise size, outbreak detection, and following notification, would not necessarily require a proportional increase in the number of fatalities with premise size. | The fatality threshold reflects the maximum scale of outbreaks that the current surveillance system fails to contain across different premise sizes. For example, a higher fatality threshold indicates a less timely detection of outbreaks, and vice versa. |

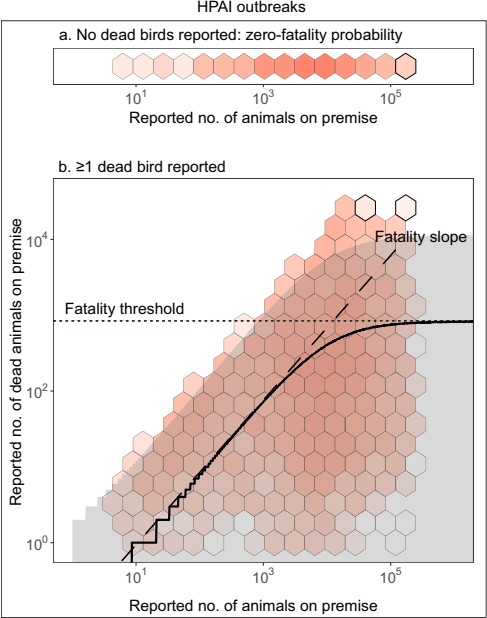
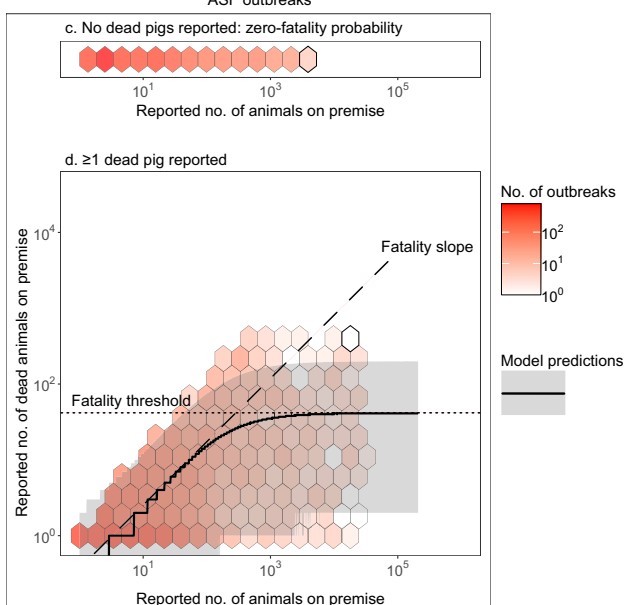

**Fig. 1 | Conceptual modelling framework for performance in HPAI and ASF outbreak detection.** The colour density of the hexagons represents the number of outbreaks included in this study amongst those reported to the WOAH WAHIS. The outbreaks reported with zero fatalities are shown at the top (**a** and **c**) by the number of susceptible animals on a premise (x-axis), while those reported with any fatalities are shown at the bottom (**b** and **d**) by the number of susceptible animals on a premise (x-axis) and the number of dead animals (y-axis). In **a** and **c**, the framework proposes that outbreaks can be detected before deaths occur, with a certain probability (estimated as zero-fatality probability). In **b** and **d**, the solid lines and grey shades represent the median and 95th percentile intervals of the number of dead animals over the number of susceptible animals on a premise for outbreaks reported with any fatalities, as predicted by the baseline models. The framework proposes that fatalities initially show a positive correlation with premise size (estimated as fatality slope, dashed lines), the extent of which decreases as the number of susceptible animals on a premise increases and eventually disappear beyond a certain number of susceptible animals on a premise (estimated as fatality threshold, dotted lines). Source data is provided as a Source Data file.

**Table 2 | Evaluation and comparison of models fitted to HPAI fatality data**

| Model ID | Contextual variables | | | | | | LOOIC | ΔELPD | \|ΔELPD/ΔSE\| | Deviance R² (%) |
|---|---|---|---|---|---|---|---|---|---|---|
| | Country/territory | Premise type[a] | Season[b] | Outbreak cluster[c] | Outbreak cluster by income status[d] | Outbreak cluster by veterinary service capacity[e] | | | | |
| Null | Each fatality metric was independent of premise size and other variables | | | | | | 78789.5 | −2686.9 | 34.6 | |
| Baseline | Each fatality metric was dependent on premise size but independent of other variables | | | | | | 76794.8 | −1689.6 | 24.6 | 15.2 |
| 1 | O | | | | | | 73564.4 | −74.3 | 3.1 | 37.7 |
| 2.1 | O | O | | | | | 73450.4 | −17.3 | 0.8 | 38.3 |
| 2.2 | O | | O | | | | 73541.4 | −62.9 | 3.9 | 38.0 |
| 2.3 | O | | | O | | | 73548.4 | −66.4 | 2.8 | 37.9 |
| 2.4 | O | | | | O | | 73540.8 | −62.6 | 2.8 | 37.9 |
| 2.5 | O | | | | | O | 73562.2 | −73.2 | 3.0 | 37.8 |
| 3.1 | O | O | O | | | | 73448.8 | −16.5 | 2.1 | 38.6 |
| 3.2 | O | O | | O | | | 73433.2 | −8.8 | 0.4 | 38.6 |
| 3.3 | O | O | | | O | | 73422.7 | −3.5 | 0.2 | 38.6 |
| 3.4 | O | O | | | | O | 73438.8 | −11.6 | 0.6 | 38.3 |
| **4** | **O** | **O** | **O** | **—** | **O** | **–** | **73415.7** | **—** | **—** | **39.1** |

The best-fitting model is highlighted in bold.

[a]Commercial farms, backyard farms, and villages (reference group: outbreaks on backyard farms).

[b]Autumn, winter, spring, and summer (reference group: outbreaks reported in autumn).

[c]Inclusion of an outbreak in country/territory-specific space-time outbreak clusters (reference group: outbreaks not in the clusters).

[d]Outbreak cluster variable differentiated by country/territory income level (reference group: outbreaks not in the clusters and in upper-middle income countries).

[e]Outbreak cluster variable accounting for veterinary service capacity, which was defined as the number of veterinarians working on animal health activities in public administrations per a million terrestrial animal biomass.

LOOIC Leave-one-out information criterion;

ΔELPD Expected log pointwise predictive density differences from the lowest LOOIC model;

ΔSE Standard error differences from the model with the lowest LOOIC model.

**Table 3 | Associations of zero-fatality probability, fatality slope, and fatality threshold with variables in the best-fitting model for HPAI outbreaks**

| Variable | No. of HPAI outbreaks (%) | Zero-fatality probability OR per (95% HDI) | Fatality slope Log-odds (95% HDI) | Fatality threshold Log-odds (95% HDI) |
|---|---|---|---|---|
| *Premise size (no. of susceptible animals)[a]* | | **0.54 (0.48 to 0.59)** | Not applicable | Not applicable |
| *Premise type* | | | | |
| Commercial farm | 5477 (87.4) | Reference group | | |
| Backyard farm/village | 789 (12.6) | **0.15 (0.08 to 0.23)** | **1.81 (1.08 to 2.57)** | **−2.42 (−3.01 to −1.51)** |
| *Season[b]* | | | | |
| Winter (Dec, Jan, Feb) | 2489 (39.7) | Reference group | | |
| Spring (Mar, Apr, May) | 2366 (37.8) | 1.17 (0.81 to 1.57) | **−0.71 (−0.98 to −0.46)** | **0.99 (0.57 to 1.38)** |
| Summer (Jun, July, Oct) | 475 (7.6) | 1.38 (0.99 to 1.85) | −0.16 (−0.55 to 0.24) | 0.08 (−0.29 to 0.52) |
| Autumn (Sep, Oct, Nov) | 936 (14.9) | **0.64 (0.46 to 0.85)** | −0.18 (−0.53 to 0.20) | 0.21 (−0.07 to 0.50) |
| *Space-time outbreak cluster by income status* | | | | |
| Not in space-time outbreak cluster | | | | |
| High-income countries | 2256 (36.0) | Reference group | | |
| Upper-middle income countries | 206 (3.3) | 0.87 (0.13 to 2.79) | 0.92 (−0.89 to 3.00) | −1.14 (−2.68 to 0.24) |
| Lower-middle income countries | 378 (6.0) | **0.10 (0.01 to 0.40)** | −0.32 (−1.82 to 1.11) | 0.39 (−1.47 to 2.51) |
| In space-time outbreak cluster | | | | |
| High-income countries | 2653 (42.3) | **0.60 (0.51 to 0.70)** | 0.06 (−0.17 to 0.30) | −0.12 (−0.39 to 0.14) |
| Upper-middle income countries | 215 (3.4) | 1.05 (0.12 to 3.60) | −0.66 (−2.34 to 1.04) | −1.14 (−2.66 to 0.34) |
| Lower-middle income countries | 558 (8.9) | 0.24 (0.02 to 1.00) | −0.42 (−1.85 to 1.05) | −0.86 (−2.46 to 0.69) |
| Total | 6266 (100.0) | | | |

Statistically significant associations are highlighted in bold.

*OR Odds ratio, 95% HDI = 95% highest density interval*

[a]The OR represents the effect of a one-unit increase in the log10 of premise size. Only for the zero-fatality probability. Premise size was accounted for by the functional relationship between the fatality slope and fatality threshold.

[b]For countries in the southern hemisphere, seasons were defined in the opposite way.

**Table 4 | Evaluation and comparison of models fitted to ASF fatality data**

| Model ID | Model variables | | | | | | LOOIC | ΔELPD | \|ΔELPD/ΔSE\| | Deviance R² (%) |
|---|---|---|---|---|---|---|---|---|---|---|
| | Country/ territory | Premise type[a] | Season[b] | Outbreak cluster[c] | Outbreak cluster by income status[d] | Outbreak cluster by veterinary service capacity[e] | | | | |
| Null | Each fatality metric was independent of premise size and other variables | | | | | | 57291.8 | −6276.6 | 56.7 | — |
| Baseline | Each fatality metric was dependent on premise size but independent of other variables | | | | | | 50261.2 | −2761.3 | 39.8 | 35.6 |
| 1 | O | | | | | | 45636.9 | −449.1 | 9.1 | 54.0 |
| 2.1 | O | O | | | | | 44869.7 | −65.5 | 4.1 | 57.7 |
| 2.2 | O | | O | | | | 45603.8 | −432.6 | 9.1 | 54.1 |
| 2.3 | O | | | O | | | 45648.4 | −454.9 | 9.1 | 53.8 |
| 2.4 | O | | | | O | | 45557.2 | −409.3 | 8.5 | 54.8 |
| 2.5 | O | | | | | O | 45642.7 | −452.0 | 9.1 | 54.0 |
| 3.1 | O | O | O | | | | 44827.8 | −44.6 | 3.3 | 57.4 |
| 3.2 | O | O | | O | | | 44861.6 | −61.5 | 3.9 | 57.3 |
| 3.3 | O | O | | | O | | 44777.7 | −19.5 | 2.0 | 57.8 |
| 3.4 | O | O | | | | O | 44872.1 | −66.7 | 4.1 | 57.5 |
| **4** | **O** | **O** | **O** | **—** | **O** | **—** | **44738.7** | **—** | **—** | **57.7** |

The best-fitting model is highlighted in bold.
[a]Commercial farms, backyard farms, and villages (reference group: outbreaks on backyard farms).
[b]Autumn, winter, spring, and summer (reference group: outbreaks reported in autumn).
[c]Inclusion of an outbreak in country/territory-specific space-time outbreak clusters (reference group: outbreaks not in the clusters).
[d]Outbreak cluster variable differentiated by country/territory income level (reference group: outbreaks not in the clusters and in upper-middle income countries).
[e]Outbreak cluster variable accounting for veterinary service capacity, which was defined as the number of veterinarians working on animal health activities in public administrations per a million terrestrial animal biomass.
LOOIC Leave-one-out information criterion;
ΔELPD Expected log pointwise predictive density differences from the lowest LOOIC model;
ΔSE Standard error differences from the model with the lowest LOOIC model.

### Association of variables with outbreak notification performance

Tables 3, 5 show the associations of premise size, premise type, season, and outbreak cluster by income status with the fatality metrics of HPAI and ASF outbreaks, respectively. Fatality metric results are presented first for HPAI, then for ASF.

For HPAI, for a log10 increase in premise size, the odds of an outbreak being notified with no deaths decreased by 46.0% (odds ratio [OR]: 0.54, 95%HDI:0.48–0.59). For premise type, the odds of zero fatalities at initial notification decreased by 85.0% (OR: 0.15, 95%HDI: 0.08–0.23) for backyard farms and villages compared to commercial farms. For season, the HPAI zero-fatality probability was the lowest in autumn, showing a significantly lower odds compared to winter (OR: 0.64, 95%HDI: 0.46–0.85). Finally, for HPAI outbreaks outside clusters, the odds of zero fatalities was significantly higher in high-income countries/territories (reference group) than in lower-middle income countries/territories (OR: 0.10, 95%HDI: 0.01–0.40). Notably, in high-income countries/territories, the odds of zero fatalities was lower within than outside outbreak clusters (OR: 0.60, 95%HDI: 0.51–0.70). However, within upper-middle and lower-middle income countries/territories, being part of clusters did not significantly change the odds of zero fatalities.

For premise type, compared to commercial farms, the log-odds of HPAI fatality slope increased by 1.81 (95%HDI: 1.08–2.57) in backyard farms and villages. Regarding season, the fatality slope was the lowest in spring for HPAI outbreaks, with its log-odds decreasing by 0.71 (95% HDI: −0.98 to −0.46) compared to winter. For outbreak cluster by income status, no significant associations with the HPAI fatality slope were found.

Additionally, compared to commercial farms, the HPAI fatality threshold was significantly lower in backyard farms and villages (log-odds change: −2.42, 95%HDI: −3.01 to −1.51). Regarding season, compared to winter, the log-odds of HPAI fatality threshold increased by 0.99 (95%HDI: 0.57–1.38) in spring. For outbreak cluster by income

status, no significant associations with the HPAI fatality threshold were found.

For ASF, for a log10 increase in premise size, the odds of an ASF outbreak being notified with no deaths decreased by 33.0% (OR: 0.67, 95%HDI: 0.61–0.73). For premise type, backyard farms had 1.88 times higher odds of zero fatalities (95%HDI: 1.26–2.70), whereas villages had 0.38 times lower odds (95%HDI: 0.09–0.84), both compared to commercial farms. No significant associations were found for season and outbreak cluster by income status.

For premise type, the log-odds of ASF fatality slope increased by 2.70 (95%HDI: 2.29–3.15) in backyard farms compared to commercial farms. Regarding season, the log-odds of ASF fatality slope decreased by 0.57 (95%HDI: −1.01 to −0.16) in summer and by 0.49 (95%HDI: −0.91 to −0.07) in autumn, compared to winter.

Compared to commercial farms, the ASF fatality threshold was significantly lower in backyard farms (log-odds change: −2.02, 95%HDI: −2.17 to −1.88). Regarding season, the log-odds of ASF fatality threshold decreased by 0.26 in summer (95%HDI: −0.40 to −0.11) and by 0.27 in autumn (95%HDI: −0.41 to −0.14) compared to winter. Finally, the ASF fatality threshold was found higher in upper-middle income countries/territories than in high-income countries/territories. For outbreaks occurring outside clusters, the log-odds of ASF fatality threshold increased by 3.09 (95%HDI: 1.72–4.62) in upper-middle income countries/territories than high-income countries/territories. For outbreaks occurring within clusters, the log-odds of ASF fatality threshold increased additionally by 3.14 (95%HDI: 1.72–4.63).

### Comparison of outbreak notification performance between countries/territories

Figures 2, 3 show by country/territory (i) the total number of fatalities recorded in the initial report for all analysed outbreaks, (ii) the number predicted by the best-fitting model, and (iii) the number predicted by a model refitted by excluding country/territory while retaining all other variables from the best-fitting model, for HPAI and ASF outbreaks,

**Table 5 | Associations of zero-fatality probability, fatality slope, and fatality threshold with variables in the best-fitting model for ASF outbreaks**

| Variable | No. of ASF outbreaks (%) | Zero-fatality probability OR (95% HDI) | Fatality slope Log-odds (95% HDI) | Fatality threshold Log-odds (95% HDI) |
|---|---|---|---|---|
| *Premise size (no. of susceptible animals)[a]* | | **0.67 (0.61 to 0.73)** | Not applicable | Not applicable |
| *Premise type* | | | | |
| Commercial farm | 1413 (13.5) | Reference group[d] | | |
| Backyard farm[c] | 7836 (74.9) | **1.88 (1.26 to 2.70)** | **2.70 (2.29 to 3.15)** | **−2.02 (−2.17 to −1.88)** |
| Village | 1212 (11.6) | **0.38 (0.09 to 0.84)** | 0.52 (−0.11 to 1.17) | 0.13 (−0.35 to 0.66) |
| *Season[b]* | | | | |
| Winter (Dec, Jan, Feb) | 1630 (15.6) | Reference group[d] | | |
| Spring (Mar, Apr, May) | 996 (9.5) | 0.82 (0.68 to 1.01) | −0.16 (−0.66 to 0.34) | −0.06 (−0.23 to 0.11) |
| Summer (Jun, July, Oct) | 3894 (37.2) | 1.03 (0.88 to 1.19) | **−0.57 (−1.01 to −0.16)** | **−0.26 (−0.40 to −0.11)** |
| Autumn (Sep, Oct, Nov)[c] | 3941 (37.7) | 1.16 (0.99 to 1.34) | **−0.49 (−0.91 to −0.07)** | **−0.27 (−0.41 to −0.14)** |
| *Space-time outbreak cluster by income status* | | | | |
| Not in space-time outbreak cluster | | | | |
| High-income countries | 363 (3.5) | Reference group[d] | | |
| Upper-middle income countries[c] | 4245 (40.6) | 2.62 (0.31 to 8.80) | −0.79 (−2.41 to 0.74) | **3.09 (1.72 to 4.62)** |
| Lower-middle income countries | 410 (3.9) | 0.32 (0.01 to 1.88) | 1.97 (−0.12 to 4.20) | 0.18 (−1.74 to 2.16) |
| In space-time outbreak cluster | | | | |
| High-income countries | 467 (4.5) | 1.17 (0.65 to 1.75) | 0.17 (−0.16 to 0.55) | −1.10 (−1.50 to −0.66) |
| Upper-middle income countries | 4013 (38.4) | 2.38 (0.32 to 7.95) | −0.90 (−2.54 to 0.79) | **3.14 (1.72 to 4.63)** |
| Lower-middle income countries | 963 (9.2) | 1.21 (0.04 to 6.75) | 1.39 (−0.77 to 3.56) | 0.48 (−1.46 to 2.43) |
| Total | 10461 | | | |

Statistically significant associations are highlighted in yellow.

*OR* Odds ratio; 95% *HDI* = 95% highest density interval.

[a]The OR represents the effect of a one-unit increase in the log10 of premise size. Only for the zero-fatality probability. Premise size was accounted for by the functional relationship between the fatality slope and fatality threshold.

[b]For countries in the southern hemisphere, seasons were defined in the opposite way.

[c]Categories set as the reference groups during Hamiltonian Monte Carlo (HMC) estimation.

[d]Categories set as the reference groups for the presentation of the results, in line with the reference groups for HPAI outbreaks.

respectively. The predicted values from the model excluding country/territory corresponded to the total number of fatalities expected under the assumption of average performance across all analysed countries/territories while accounting for the other variables. Figures S3–S7 present this information by country/territory and premise type. Additionally, for countries/territories estimated to have performed significantly better (i.e., experiencing significantly fewer fatalities) or significantly worse (i.e., experiencing significantly more fatalities), we reviewed their fatality metric estimates to better understand the underlying drivers of their surveillance performance (Figs. S8–S12). For both HPAI and ASF outbreaks, the total number of fatalities predicted by the best-fitting model closely matched the number recorded in the initial report, except for ASF outbreaks from one country/territory (ID: 37) (Figs. 2, 3). The outbreaks from this country/territory were all notified from villages and were estimated to have exceptionally high zero-fatality probability and fatality slope estimates, compared to village outbreaks in other countries/territories (Fig. S12).

The countries/territories that performed significantly better tended to have a high zero-fatality probability as well as low fatality slope and threshold estimates (e.g., ID: 33 for both HPAI and ASF outbreaks) (Figs. S8, S10). In contrast, those that performed significantly worse exhibited the opposite trends, with a low zero-fatality probability (Figs. S8–S12). In most countries/territories, for both HPAI and ASF outbreaks, the fatality slope was estimated to be high amongst backyard farms, compared to commercial farms (Figs. S9, S11). Pathogen-specific comparisons are described below.

For HPAI, six countries (17.1%, ID: 7, 8, 14, 30, 31, and 33) of the 35 analysed countries/territories were estimated to have experienced significantly fewer fatalities than expected assuming the average performance (blue background shading in Fig. 2). In contrast, four countries/territories (11.4%, ID: 10, 16, 21, and 27) were estimated to have experienced significantly more fatalities (red background shading in Fig. 2). The analysis of country/territory-level performance, separately for each premise type, showed that these significant differences were driven by HPAI outbreaks on commercial farms, rather than backyard farms and villages (Figs. S3, S4).

For ASF, six (23.0%, ID: 2, 17, 23, 25, 33, and 37) of the 26 analysed countries/territories were estimated to have performed better with significantly fewer fatalities (blue background shading in Fig. 3), and four countries/territories (15.4%, ID: 34, 36, 41, and 46) to have performed worse with significantly more fatalities (red background shading in Fig. 3) than expected assuming the average performance. When analysed separately for each premise type, countries/territories that performed significantly better or worse were identified across different premise types, suggesting that both premise types were responsible for the pattern observed when they were analysed together, contrasting to HPAI outbreaks (Figs. S5–S7).

## Discussion

The results of our analysis, drawn from WOAH WAHIS outbreak notification data, offer valuable insights into the variability of surveillance performance in HPAI and ASF outbreak notification within and between countries/territories, and the key factors associated with this variability.

While several attributes have been developed to evaluate animal infectious disease surveillance systems, assessing these attributes has been constrained by a lack of standardised data, as well as the absence

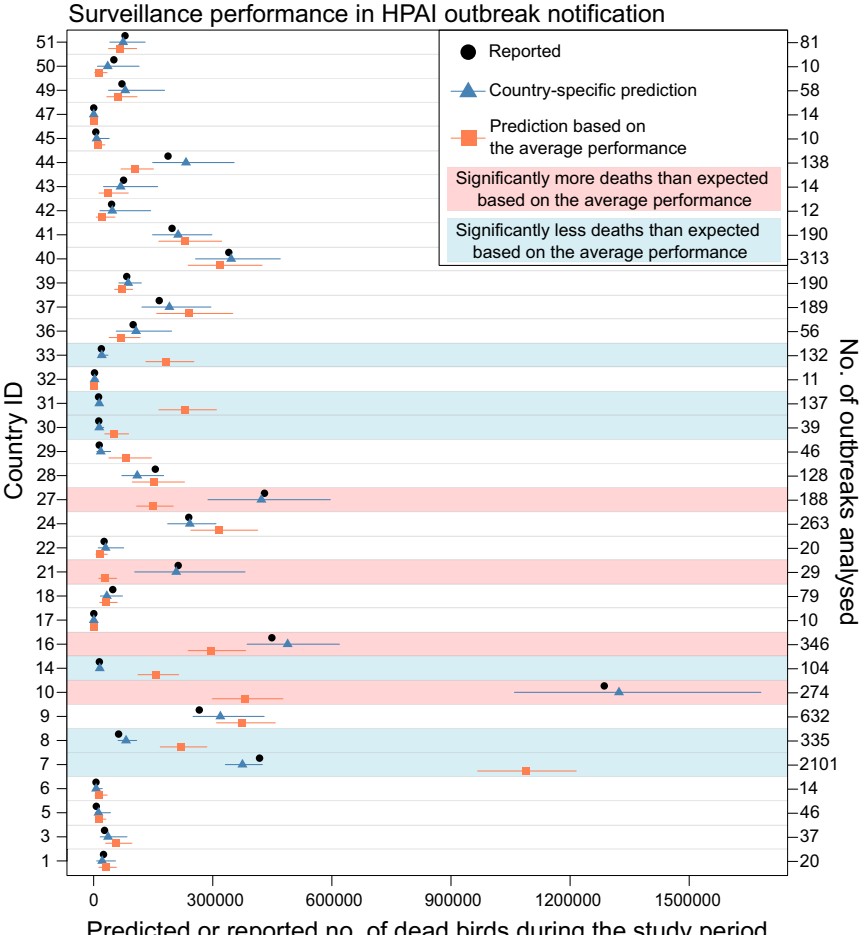

**Fig. 2 | Country/territory-level performance in HPAI outbreak notification.** The black points represent the total number of deaths across all analysed outbreaks (*x*-axis), as recorded at initial notification, by country/territory (IDs on the left *y*-axis and the number of outbreaks analysed on the right *y*-axis). The blue triangles represent the median number of deaths predicted by the best-fitting model, while the orange squares represent the median number predicted by a model excluding the country variable but including other variables from the best-fitting model. The corresponding horizontal lines represent the 95th percentile range. The blue background shading indicates countries estimated to have performed significantly better than average (i.e., the upper limit of the 95th percentile intervals [PI] predicted by the best-fitting model <lower limit of the 95th PI predicted by the model excluding country/territory). The red background shading indicates those estimated to have performed significantly worse than average (i.e., the lower limit of the 95th PI predicted by the best-fitting model > upper limit of the 95th PI predicted by the model excluding country/territory). Source data is provided as a Source Data file.

of frameworks and tools tailored to analyse such data[7,8,12,13]. Therefore, our framework provides unique insights into the evaluation of animal infectious disease surveillance systems by incorporating three fatality metrics that explicitly account for the outbreak progression expected following pathogen introduction until notification. Furthermore, using outbreak notification data collected under WOAH guidelines within our framework enables a systematic evaluation of surveillance performance across countries/territories, which is crucial for controlling transboundary infectious diseases. This approach can also be applied at sub-country/territory levels, over time, and potentially to other diseases, and its robustness could be further improved through ongoing efforts to standardise such data.

In our modelling framework, each fatality metric has distinct—though not mutually exclusive—implications for surveillance performance. However, we emphasise that these fatality metrics should be assessed in light of pathogen characteristics: HPAI viruses typically spread rapidly with short latent period and generation time, leading to high fatalities within a short period[14], whereas ASF viruses spread more slowly, with a more gradual onset of clinical signs and deaths[15]. Importantly, data completeness, accuracy, and bias—commonly considered attributes of surveillance evaluation[8,12,13]—require continuous

assessment. Additionally, while our study is a foundational step in assessing the variability of surveillance performance, follow-up studies on eco-social, veterinary, and agricultural factors are needed to better understand the observed variability and develop country-/territory-specific strategies.

For both HPAI and ASF, the number of susceptible animals and the country/territory of an outbreak were identified as the most significant factors. Regarding the number of susceptible animals, this result was expected as pathogen transmission likely increases with larger susceptible populations, and it was why we constructed the baseline model with the number of susceptible animals as a key variable, assuming primarily density-dependent transmission[16]. More specifically, the zero-fatality probability was estimated to decrease as the number of susceptible animals increased, suggesting that early detection becomes more challenging in larger herds, likely due to factors including higher transmission risk, background mortality, and husbandry conditions. However, beyond a certain population size, fatalities would not increase linearly with a larger number of susceptible animals when surveillance is in place, and our modelling framework effectively accounted for and demonstrated this dynamic.

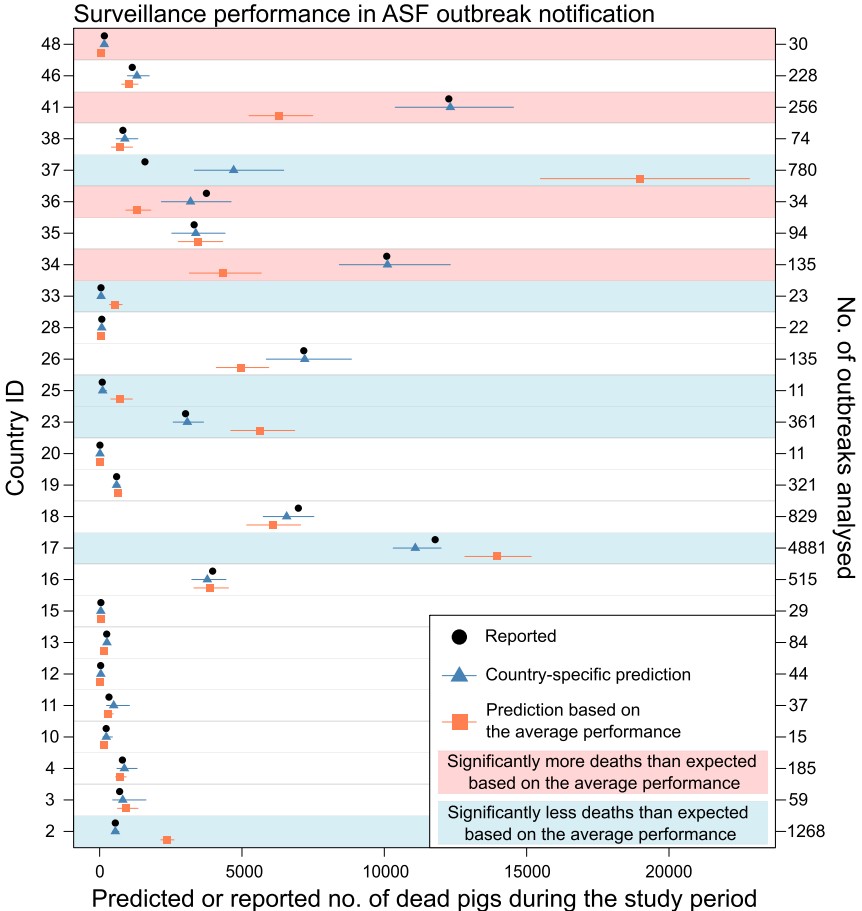

**Fig. 3 | Country/territory-level performance in ASF outbreak notification.** The black points represent the total number of deaths across all analysed outbreaks (x-axis), as recorded at initial notification, by country/territory (IDs on the left y-axis and the number of outbreaks analysed on the right y-axis). The blue triangles represent the median number of deaths predicted by the best-fitting model, while the orange squares represent the median number predicted by a model excluding the country variable but including other variables from the best-fitting model. The corresponding horizontal lines represent the 95th percentile range. The blue background shading indicates countries estimated to have performed significantly better than average (i.e., the upper limit of the 95th percentile intervals [PI] predicted by the best-fitting model < lower limit of the 95th PI predicted by the model excluding country/territory). The red background shading indicates those estimated to have performed significantly worse than average (i.e., the lower limit of the 95th PI predicted by the best-fitting model > upper limit of the 95th PI predicted by the model excluding country/territory). Source data is provided as a Source Data file.

The results regarding the country/territory of an outbreak indicate that the number of fatalities reported at initial notification is highly heterogeneous across countries/territories. Such heterogeneity could stem from a variety of country/territory-level factors not considered in our analysis, including the capacity of veterinary services and the level of viral circulation within countries/territories. In our analysis, a few countries/territories were estimated to have experienced significantly more or fewer fatalities than expected assuming the average performance across all analysed countries/territories. For these countries/territories, rather than simply attempting to transplant practices deemed successful in one setting to another, gathering detailed information on the eco-social, veterinary, and agricultural factors and assessing it in relation to differences in fatality metrics could help identify the key drivers of surveillance performance. In this process, it is important to keep in mind that the fatality metrics capture only a limited dimension of animal health systems and therefore cannot be used as the sole measure of their overall performance. This approach would then help understand how different contexts influence the actual operation of surveillance, providing a critical foundation for developing context-specific strategies to strengthen animal infectious disease surveillance at national, regional, and global levels.

Additionally, this assessment should also include scrutiny of potential differences in reporting practices and accuracy. The only country/territory (ID: 37) for which the best-fitting model did not accurately predict the number of fatalities from ASF outbreaks in villages serves as a good example. This country/territory was estimated to perform significantly better than expected based on the average performance. However, it had an exceptionally high zero-fatality probability compared to other countries/territories and also exhibited a high fatality slope. This pattern contrasted with other countries/territories also estimated to perform significantly better, since they tended to be associated with both a high zero-fatality probability and low fatality slope and threshold estimates, like country/territory 33 for both HPAI and ASF outbreaks. This suggests that the outlier behaviour of the country/territory 37 warrants further investigation to ensure that the observed trends genuinely reflect surveillance capacity or characteristics rather than being artefacts of data reporting issues. Such investigations could therefore begin by assessing potential systematic differences in reporting practices, for example, how village outbreaks were defined and reported. Although outbreaks were notified using a standardised format, there is no consensus on the definition of the premise types, which may lead to inconsistencies in classification, particularly between outbreaks in backyard farms and

villages. If no significant differences are identified, the investigation could then focus on the epidemiological context of these villages and how surveillance systems operated there, compared to those in other countries/territories.

After controlling for the number of susceptible animals and the country/territory of an outbreak, other variables, including premise type, season, and the cluster occurrence of outbreaks, were significantly associated with the number of fatalities reported at initial notification. However, it is important to note that these factors did not contribute much to explaining the overall variability in the data, potentially due to the large size of the dataset: a large statistical power may have allowed detection of relatively small effects. That is, although these variables were found to have statistically significant associations, the practical relevance of their impact on the number of fatalities appears to be less important than that of the number of susceptible animals and the country/territory of an outbreak.

Regarding premise type, HPAI outbreaks on backyard farms and in villages were associated with a lower zero-fatality probability and a higher fatality slope compared to those on commercial farms. This pattern likely stems from a combination of relatively poor biosecurity and surveillance measures in these settings, as well as the inherent difficulty of early detection in HPAI outbreaks. The virus's rapid spread and high lethality could lead to fatalities before detection through other clinical signs or active surveillance is possible, particularly in backyard and village settings. On the other hand, ASF outbreaks on backyard farms, while showing a higher fatality slope, were significantly more likely to be reported with zero fatalities. However, this does not necessarily indicate that backyard farms are inherently better at early detection of ASF outbreaks, given their typically poorer surveillance compared to commercial farms[17,18]. A more plausible explanation is that surveillance efforts, when present, were intensified toward backyard farms due to their known susceptibility to ASF virus introduction, particularly from infected wild boar populations[19]. This, combined with the relatively slow progression of ASF outbreaks, may have made early detection easier.

Regarding season, our analysis revealed that the probability of reporting zero fatalities at initial notification was lowest in autumn for HPAI outbreaks and in spring for ASF outbreaks. Notably, these seasons occur just before the major transmission periods—winter for HPAI[4] and summer/autumn for ASF[10]. One possible explanation is that farmers' risk awareness, and potentially surveillance sensitivity, might have been lower before these major transmission seasons. Conversely, the seasons following the major transmission periods (that is, spring for HPAI and winter for ASF) showed similar levels of zero-fatality reporting as during the peak transmission seasons. This suggests that the enhanced surveillance and heightened risk awareness established during the high-risk periods likely persisted into the subsequent season. On the other hand, the fatality slope and fatality threshold exhibited notable variations in several, but not all, seasons, compared to the high-risk periods for both HPAI and ASF outbreaks. When outbreaks advance to the point of causing deaths, detection of HPAI and ASF outbreaks may primarily rely on noticeable changes in overall mortality compared to expected background mortality, along with other clinical signs. Therefore, seasonal fluctuations in background mortality and possibly population sizes could have influenced the fatality slope and fatality threshold[20–22]. However, our results reflect the average trends across the countries/territories analysed, and therefore, different countries/territories may exhibit unique trends due to different climates and animal production cycles.

In lower-middle income countries/territories, the lower zero-fatality probability for HPAI outbreaks occurring outside of clusters suggests that, compared to high-income countries/territories, the ability to detect these outbreaks early—before fatalities occurred—was already limited, even during periods of lower outbreak intensity. In high-income countries/territories, signs of surveillance burden were evident during periods of intense outbreak activity, as indicated by the lower zero-fatality probability for HPAI outbreaks occurring within clusters. For upper-middle and lower-middle income countries/territories, outbreaks within clusters did not show a significant decrease in the zero-fatality probability, likely due to an already low zero-probability during periods of lower outbreak intensity. In contrast, for ASF outbreaks, the zero-fatality probability did not differ significantly between clustered and non-clustered outbreaks, suggesting that early detection of ASF outbreaks was relatively consistent, regardless of their occurrence within or outside clusters. On the other hand, compared to high-income countries/territories, ASF outbreaks in upper-middle-income countries/territories, both within and outside clusters, exhibited a significantly higher fatality threshold. ASF outbreak detection may have been delayed in these countries/territories during intense outbreak periods, leading to larger fatalities reported at notification.

It is important to note that outbreak clusters were identified based on both HPAI and ASF outbreaks, under the assumption that they exert equal pressure on veterinary authorities. However, the pressure exerted by HPAI and ASF outbreaks, as well as other pathogens, on veterinary authorities likely varies between countries/territories depending on multiple factors, including their surveillance and control capacities. We therefore used 180 days as the maximum temporal window to allow for the detection of all relevant clusters within the study period (e.g., short, intense clusters as well as long, protracted clusters), while still considering seasonal HPAI and ASF transmission patterns. Shorter maximum windows might fail to capture long, protracted clusters that place significant pressure on veterinary services.

For both HPAI and ASF outbreaks, the best-fitting model indicated high overdispersion in fatalities at initial outbreak notification, after accounting for the variables included. This suggests that the performance in outbreak notification is highly variable amongst premises within the same countries/territories. While this could arise from the stochasticity of viral transmission within premises (e.g., fatality patterns could be more heterogeneous depending on the course of viral transmission and the route of viral introduction), other important factors not considered in this study could also have played a role behind this variability, including differences in the level of on-premise biosecurity and surveillance, animal density, host breeds and species, transmissibility and virulence by viral types, and surveillance capacity between regional veterinary services. These emphasise the need to identify critical areas to improve surveillance capacity within as well as across countries/territories.

Finally, although our modelling framework was not specifically designed to compare differences in the outbreak dynamics between the HPAI and ASF viruses, certain observations suggest important surveillance implications arising from these differences. Compared to commercial farms, ASF outbreaks on backyard farms were associated with a higher zero-fatality probability, while HPAI outbreaks on backyard and village settings were linked to a lower zero-fatality probability. As discussed earlier, this high zero-fatality probability for backyard ASF outbreaks may have been influenced by surveillance efforts targeted at this setting. However, despite this, detection of ASF outbreaks before observing fatalities would likely be possible largely due to the relatively slow progression of ASF[15]. In contrast, HPAI outbreaks are characterised by rapid viral spread and resulting deaths[14], which can be exacerbated by a lack of biosecurity in backyard and village settings, making early detection particularly challenging. From a surveillance perspective, this suggests that HPAI outbreaks, despite causing large fatalities, are relatively easier and faster to detect compared to ASF outbreaks, potentially leading to better surveillance sensitivity and timeliness. The slow progression of ASF outbreaks, on the other hand, allows for extensive viral circulation within premises and provides a larger window of opportunity for the virus to spread to other pig populations by the time mortality levels rise enough to

trigger detection. This factor needs to be considered when improving ASF outbreak detection sensitivity and timeliness.

This study has limitations. Firstly, while the study period for HPAI outbreaks was designed to cover the global spread of wild bird-adapted HPAI H5N1 viruses, other HPAI viruses also circulated concurrently, affecting different regions. These other subtypes may have contributed to the observed heterogeneities in the number of fatalities due to their varying levels of transmissibility and virulence.

Secondly, the host animals in HPAI outbreaks were categorised as birds in the WOAH WAHIS. HPAI viruses, however, are known to pose varying levels of transmissibility and virulence to different bird species. Notably, chickens are generally more susceptible than ducks to HPAI viruses[23,24]. This suggests that our fatality metrics could be influenced by the composition of bird species within a given country. In chicken farms, a sudden rise in mortality can serve as an early warning sign of an outbreak, but by the time it is detected, substantial losses may have already occurred, potentially leading to a low zero-fatality probability and a high fatality slope. In contrast, HPAI outbreaks in duck farms may spread widely before detection with a few or no deaths, potentially resulting in a high zero-fatality probability and a low fatality slope. Therefore, having information on host species could help improve the assessment of surveillance performance in HPAI outbreak notification, ensuring that differences in species susceptibility and transmissibility are accounted for. For ASF outbreaks, accounting for viral genotypes and swine breeds would also allow more robust parameter estimation, although these concerns may be less relevant compared to the HPAI host/viruses, given the relatively stable viral genetic structure and the less variable transmissibility and virulence across different swine breeds[25,26].

Thirdly, surveillance capacity could have changed dramatically during the study periods within countries/territories as the level of viral circulation varied, veterinary services gained experience with newly circulating viruses, and socioeconomic and geopolitical contexts evolved. Such changes may have occurred in either direction, making it difficult to incorporate into our framework given limited data availability. Although we used the number of veterinarians per a million terrestrial animal biomass[27] as a proxy for veterinary service capacity, this metric does not reflect potential temporal shifts in the reporting behaviour of key stakeholders, including veterinarians, farmers, and other value chain actors. Therefore, future studies should aim to complement quantitative data with detailed descriptions of the contextual factors, such as politics, socioeconomic, and veterinary system dynamics, that may have influenced these behaviours and, ultimately, surveillance performance[28].

Fourth, while WOAH member countries/territories are required to notify outbreaks of listed diseases in a standardised format, gaps in the data existed due to variations in how outbreaks were reported. For example, in some countries/territories, the number of dead animals was not available for a relatively large proportion of ASF and, particularly, HPAI outbreaks. This could reflect differences in reporting practices within countries/territories or pressure on veterinary services during certain periods (e.g., insufficient resources to collect detailed outbreak data when they occur in clusters). The extent of this missing information was significant in a few countries/territories, and its impact would depend on the degree to which the outbreaks excluded from the analysis due to missing data had different notification dynamics. Additionally, although not widespread, there were instances where the number of dead animals was reported as greater than the number of susceptible animals or equal to the number of culled animals in several outbreaks, making it challenging to clearly interpret these numbers. These data inconsistencies underscore the need for efforts to further improve and standardise reporting practices, as well as more precise definitions of the variables to be collected from notification reports.

Considering the limitations discussed above, future research would benefit from incorporating interviews with individual veterinary services to better understand differences in surveillance and reporting practices that may introduce biases into the data. These interviews could also help obtain more detailed outbreak notification data—such as host species, viral lineages, and other key epidemiological variables, including temporal changes in surveillance capacity—collected in a way that allows for meaningful comparisons within and between countries or territories. Not only would such efforts enable more robust comparisons between countries/territories, but they could also facilitate the application of our framework at the sub-national level, helping to identify heterogeneities within countries/territories and develop more tailored, context-specific recommendations. Crucially, these activities should be conducted in close collaboration with the veterinary services involved to ensure accurate interpretation of findings and to establish a constructive feedback loop that supports the continuous strengthening of surveillance and response systems.

In conclusion, our results highlight significant variability in animal health surveillance capacities for HPAI and ASF outbreaks, both within and between countries/territories, based on WOAH WAHIS outbreak notification data. These disparities may arise from specific drivers shaped by diverse eco-social, veterinary, and agricultural contexts across countries/territories, which were not fully captured by the macro-level variables used in this study. A key next step would thus be to focus global and regional efforts on identifying the specific drivers that influence the functioning of HPAI, ASF, and other animal disease surveillance systems. By doing so, we can develop more targeted, effective, and efficient recommendations to improve animal health surveillance capacities.

## Methods
### Data
The Veterinary Authorities of WOAH Member Countries are required to notify the WOAH of infections with HPAI and ASF viruses, amongst other listed diseases, within 24 h of their (re)occurrence, according to the criteria established in chapter 1.1.3 of the Terrestrial Animal Health Code. Following the first notification of a unique epidemiological event, the Veterinary Authorities should also "provide further information on the evolution of the event which justified the notification" through weekly reports "until the listed disease has been eradicated, or the situation has become sufficiently stable". These notification reports contain various pieces of information that help assess the epidemiological context of individual outbreaks.

Amongst HPAI and ASF outbreaks reported to the WOAH through the WAHIS, this study included those that met the following selection criteria. Firstly, this study focused on outbreaks reported within specific time frames that marked significant milestones for the recent global spread of each pathogen: for HPAI, from January 1, 2020 to December 15, 2023, corresponding to the spread of wild bird-adapted HPAI H5N1 viruses; and for ASF, from January 1, 2016 to December 15, 2023, corresponding to the spread of the ASF virus in Eastern Europe, Russia, Asia, and the Americas. Secondly, for HPAI, outbreaks categorised as "High pathogenicity avian influenza viruses (poultry) (Inf. with)" by disease name and "Birds" by host species were included. For ASF, outbreaks where the disease name was "African swine fever virus (Inf. with)" and the host species name was "Swine" were included.

For each selected outbreak, data on the number of dead animals reported at initial notification, the premise size and type, and the country/territory of outbreaks were extracted from the WOAH WAHIS. When this information was not available in the initial notification report, it was obtained from the follow-up reports, if present. The outbreaks with missing information were excluded from the analysis (see Tables S1, S2 for the proportion of these outbreaks excluded by country/territory).

At the country/territory-level, outbreaks were classified as occurring in high-income, upper-high-income, upper-low-income, or low-income countries/territories, according to the World Bank Group classification. Additionally, the countries/territories' veterinary service capacity was measured as the WOAH data on the number of veterinarians working on animal health activities in public administrations per a million terrestrial animal biomass[27].

## Identification of outbreak clusters

For each country/territory, the clusters of outbreaks were identified by using a retrospective space-time permutation model[29] in SatScan (https://www.satscan.org). The space-time permutation model identified a group of outbreaks that occurred in a specific time period and geographical area significantly more frequently than what would have been expected if these outbreaks had no space-time interaction (i.e., against null space-time distributions generated through random permutations of outbreak locations and notification dates)[29]. Since the occurrence of both HPAI and ASF outbreaks can put pressure on veterinary authorities, outbreaks of both diseases were analysed together to identify spatiotemporal clusters in each country and territory. However, the outbreaks of other diseases were not considered. Clusters were identified using a monthly temporal window unit and a maximum temporal window of 6 months. This maximum window size was chosen to allow for the identification of outbreak clusters of varying sizes in space and time, while also considering the seasonal characteristics of HPAI and ASF outbreaks[10,11]. A circular spatial window was used, with the maximum possible size (i.e., the number of outbreaks within each cluster) set to 50% of all analysed outbreaks. Statistically significant clusters were detected with 999 Monte Carlo simulations, and clusters with *p-value* < 0.05 were considered statistically significant and were used to determine whether individual outbreaks occurred in clusters with other outbreaks in time and space.

The data extracted from the above steps were divided into distinct datasets based on premise type—namely, commercial farm, backyard farm, or village. In the data that met the above selection criteria, only two countries/territories reported HPAI outbreaks in villages. Considering villages and backyard farms likely share similar farming (e.g., small-scale and free roaming birds) and surveillance (e.g., limited veterinary infrastructure and informal surveillance systems) practices for poultry, in contrast to those for pigs, HPAI outbreaks in backyard farms and villages were combined, while ASF outbreaks were kept separate. Finally, to ensure sufficient data for parameter estimation, each country/territory was included only if the total number of outbreaks reported during the study period is equal to or greater than 10.

## Model

Bayesian mathematical models were fitted to $Y_i$: the number of dead animals recorded in the initial report of outbreaks $i$ notified to the WOAH-WAHIS. $Y_i$ was assumed to follow the distribution $f(X_i)$, with $X_i$ representing the predictors modulating the mean and variance of the distribution.

Formally, we used a zero-inflated negative binomial (ZINB) distribution such that:

$$Y_i \sim f(X_i) \text{ with } f : \begin{cases} P(Y_i = 0) = \delta_i \\ P(Y_i = x) \sim NB(\mu_i; \psi) \text{ for } x > 0 \end{cases} \quad (1)$$

$\delta_i$ is the probability that an outbreak was notified prior to the occurrence of any deaths, termed the 'zero-fatality probability'. $\mu_i$ is the mean number of deaths reported at notification, and $\psi$ is the overdispersion in the number of deaths reported at notification, such that the variance, $\text{Var}(Y_i)$, is given by $\text{Var}(Y_i) = \mu_i + \frac{\mu_i^2}{\psi}$. While $\delta_i$ and $\mu_i$ were assumed outbreak specific, associated with predictors $X_i$ (see

below), overdispersion ($\psi$) was assumed constant for all outbreaks after accounting for predictors $X_i$.

In a null model, a single value of $\mu$ was assumed for all outbreaks. Then, we defined a baseline model where the mean number of deaths reported at the initial notification of outbreak $i$ ($\mu_i$) was assumed to increase with the number of susceptible animals on the premise ($\alpha_i$), following the Michaelis−Menten kinetics[30]:

$$\mu_i = \frac{V_i \alpha_i}{K_m + \alpha_i} \quad (2)$$

(2) was rearranged to allow the estimation of 'fatality slope' ($\theta_i$) and 'fatality threshold' ($V_i$) as follows:

$$\alpha_i \to 0, \mu_i \approx \frac{V_i}{K_m} \alpha_i = \theta_i \alpha_i \quad (3)$$

Note that we re-parameterised the Michaelis−Menten kinetics with $\theta_i = \frac{V_i}{K_m}$ in order to estimate an epidemiologically more meaningful parameter $\theta_i$, rather than $K_m$, which in the original formulation represents the number of susceptible animals that gives rise to half the threshold:

$$\mu_i = \frac{\theta_i \alpha_i}{1 + \frac{\theta_i \alpha_i}{V_i}} \quad (4)$$

Here, the 'fatality slope' ($\theta_i$) and the 'fatality threshold' ($V_i$) described how fast the number of dead animals initially increases proportionally with the number of susceptible animals, following $\theta_i$, before reaching a plateau at $V_i$.

In the baseline model, we assumed each of the fatality metrics ($\delta$, $\theta$, and V) depended on the number of susceptible animals (i.e., premise size) for all outbreaks; the zero-fatality probability was assumed to have a log-linear relationship with premise size, while both the fatality slope and fatality threshold already incorporated premise size by nature of the model above.

Building upon the baseline model, the fatality metrics were modelled as log-linear combinations of variables representing the epidemiological contexts of individual outbreaks, including country/territory, premise type, season, outbreak cluster (with or without differentiation by country/territory income status or accounting for quantitative veterinary service capacity). Country/territory was included in all non-baseline models, and to reduce the number of models to be evaluated, the variables included were assumed to affect all three fatality metrics ($\delta$, $\theta$, and V), with their impacts estimated by separate parameters.

The relationships between these variables and fatality metrics were established using an inverse logit transformation to account for the scale of these metrics. After the transformation, no further adjustments were necessary for the zero-fatality probability and fatality slope, as both range between 0 and 1. After an inverse logit transformation, the fatality threshold also ranged between 0 and 1. Therefore, the fatality threshold was adjusted by multiplying it by the maximum observed fatality to ensure that it remained on its original scale while not exceeding its maximum possible value. When the association with outbreak cluster was assumed to differ by country/territory income status (outbreak cluster by income status), separate parameters for outbreak cluster were estimated for each income status. Conversely, when the association with outbreak cluster was assumed to vary with quantitative veterinary service capacity (outbreak cluster by veterinary service capacity), the parameter for outbreak cluster was multiplied by the number of veterinarians per a million terrestrial animal biomass as a continuous proxy measure before applying an inverse logit transformation. For each variable, the

category with most outbreaks notified was selected as the reference group. Country/territory names were anonymised.

The models were fitted within an HMC Bayesian inference framework using the RStan package[31] in R version 4.3.2[32]. The HMC sampling was iterated with four chains until convergence was considered achieved, based on visual inspection of the trace plots, Gelman-Rubin convergence diagnostic (<1.01), and the number of effective sample size (>1000). Weakly informed priors were used.

The best-fitting models were selected through a stepwise forward selection process. Briefly, premise size for the zero-fatality probability and country/territory were forced to remain in all models, as they were key variables for the model structure and study hypothesis. Other variables were added one at a time, with the variable resulting in the greatest LOOIC reduction being selected. Statistical significance between models was determined based on their differences in expected log pointwise predictive density and standard error. Additionally, the Deviance R-squared was estimated for each model, accounting for over-dispersed count data, to assess how much variability in the data was explained by the model[33].

From the best-fitting model, the association between the fatality metrics and contextual variables were assessed. Since zero-fatality probability was a probability metric, its association with contextual variables were expressed as odds ratios. In contrast, the associations of fatality slope and fatality threshold with contextual variables were expressed as log-odds changes, as these metrics were not probability metrics.

Posterior predictive cheques were conducted for the best-fitting HPAI and ASF models. In each iteration, a set of joint posterior parameter estimates was sampled. For each outbreak, posterior parameters were derived from this set, considering the variables included in the best-fitting model. The presence of dead animals for each outbreak was then simulated using the posterior zero-fatality probability. If fatalities were predicted, the number of dead animals was simulated using the posterior mean and the overdispersion of a negative binomial distribution. This process was repeated 1000 times for each outbreak, and the total predicted number of dead animals across all outbreaks was compared with the reported value for each country/territory.

Finally, a country/territory's overall performance in outbreak notification was assessed by comparing its total number of dead animals predicted by the best-fitting model with that predicted by a model in which the country/territory variable was excluded, while all other variables of the best-fitting model were retained. Therefore, while the best-fitting model estimated countries/territories' performance considering country/territory-specific parameters, the model without country/territory estimated the performance assuming no country/territory-specific effects. For instance, if the number of deaths predicted by the best-fitting model was lower (or higher) than that by the model excluding country/territory, the country/territory was considered to have performed better (or worse) than the average performance across the countries/territories, accounting for other variables included.

### Reporting summary
Further information on research design is available in the Nature Portfolio Reporting Summary linked to this article.

## Data availability
WOAH WAHIS outbreak notification data are publicly available here: https://wahis.woah.org/#/home. Source data are provided with this paper.

## Code availability
Model codes[34] are available from https://github.com/KimYounjung/WOAHOutbreakSurveillance/.

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

## Acknowledgements

P.N. and Y.K. acknowledge support from the BBSRC, through the ERA-NET ICRAD programme (BB/V019945/1).

## Author contributions

Y.K. and P.N. conceived the study. Y.K. and P.N. developed the conceptual framework and performed the analyses. P.T. and G.T. provided WOAH WAHIS outbreak notification data and detailed information on data collection and collation. G.F., P.T., G.T. and D.P. advised on animal health surveillance systems in relation to the analyses. G.F. and R.M. reviewed and advised on the conceptual modelling framework and the analyses. Y.K. wrote the first draft of the manuscript. All authors contributed to manuscript revision, read, and approved the final manuscript.

## Competing interests

The authors declare no competing interests.
