## [Transparent Peer Review file · Nature Communications]

Evaluation of Global Outbreak Surveillance Performance for High Pathogenicity Avian Influenza and African Swine Fever

Corresponding Author: Dr Younjung Kim

Version 0:

Reviewer comments:

Reviewer #1

(Remarks to the Author)

Review of the manuscript 'Surveillance Performance in HPAI and ASF Outbreak Notification: Modelling Insights from WOAHS WAHIS Notification Data' by Younjung Kim and colleagues, submitted to Nature Communications.

Summary

In this manuscript, the authors present a modelling framework to assess the performance of surveillance systems in reporting high pathogenicity avian influenza (HPAI) and African swine fever (ASF) outbreaks within and between member countries and territories of the World Organisation for Animal Health (WOAH). Data for this study were obtained from outbreaks reported between January 2020 and March 2023 (HPAI) and January 2016 to March 2023 (ASF). The model estimates three parameters, which are referred to as the 'fatality metrics': a zero-fatality probability, fatality slope, and fatality threshold. The models with the best fit for HPAI and ASF WAHIS outbreak data include the variables premise size and country/territory, as well as premise type, outbreak cluster by income status, and season, which contributed with marginal increases in the deviance explained. Thus, premise size and the country/territory were considered important factors for the performance level of surveillance systems.

General comments/evaluation

The topic of the study is highly relevant because of the increasing spillover threat of zoonotic HPAI H5N1. Many papers highlight the importance of infectious disease surveillance system strengthening, as part of 'pandemic preparedness' strategies. This manuscript aims to add to this goal using a modelling approach that assesses the performance of surveillance systems in countries reporting to the WOAHS. To this end, it uses the fatalities reported at initial outbreak notification as a performance measure. It was certainly an interesting and insightful read. However, we believe a more solution-oriented results section could have contributed to the value of this manuscript. The authors outline the factors (mainly premise size and country/territory) that play an important role in the performance of a surveillance system. However, no clear distinction is made between the surveillance strategies of countries that perform notably better or worse than average (Figures 2 and 3). If available, practical suggestions for improving surveillance in regions that score lower could add significant value to the manuscript.

The manuscript focuses on both the surveillance performance in HPAI as well as ASF outbreak performance. Both diseases have a significant impact on their susceptible hosts as well as on the economy. However, HPAI and ASF typically do not share host species, therefore, combining them does not provide additional insights in assessing the performance of the surveillance systems for each infectious disease separately. It does, however, sometimes make the text a bit more difficult to follow, or it sometimes results in the use of references which are only applicable to one of the two diseases. To improve the flow of the manuscript, or to prevent confusion, the work on both HPAI and ASF could be for instance be described in separate sub-chapters. This applies in particular to the Results section, which we believe could profit from main structuring by pathogen.

Additionally, we believe that the steps used to define clusters using SaTScan analysis could and should have been explained better, and also should be presented in the Results section. How do the results depend on this analysis? In particular, is a time window of 180 days for both ASF and HPAI plausible given the differences in the outbreak dynamics of these pathogens? Are alternative maximum radiuses considered? Can you stratify the data by cluster versus non-cluster (e.g., in Figures S1 and S2)?

On another note, it is important to take the susceptibility of different bird species for HPAI into account when scoring the surveillance performance on fatality counts. While the authors do mention the potential effect of host breeds and species (lines 341-346) on the variability in virus susceptibility – and thus, performance of outbreak notification within the same country/territory – they did not include the difference in host susceptibility in their model. Where both ducks and poultry are included in ‘birds’, HPAI appears to be more fatal to poultry than ducks. This could perhaps lead to lower zero-fatality odds for notifications from poultry farms compared to duck farms. In countries/territories where more poultry than ducks are kept, this factor might influence the overall performance level of the surveillance applied.

Overall, we found no errors in the logic or statistical analyses in this study and believe the results hold significant public health implications. However, the anonymization of countries (identified only by numbers) prevents detailed analyses and specific recommendations for individual countries. We defer to the editors to determine if this is acceptable for Nature Communications.

Specific comments

Line 37. The reference (1) is used to describe both HPAI and ASF, but it only discusses ASF. It might be helpful to find a similar reference specifically for HPAI.

Line 76-78. “Secondly, ... of the exposed population”. It is stated that larger susceptible host populations are likely to experience more fatalities due to the size of the exposed population. Are fatalities being measured in absolute numbers or proportions? For instance, while the absolute number of fatalities may be higher in larger populations than smaller populations, the proportion of fatalities is likely to be lower compared to smaller populations.

Line 99-101. “Finally, ... relative to others.” It would be helpful if the authors could clarify what is meant by ‘others.’ Were the comparisons made between different countries/territories, or between various groups?

Line 103-109. We believe a more complete description of the data is in order, together with the results of the SaTScan analysis.

Line 103-106. “The analysis ... (Figs. S1 and S2)”. The x-axes of the figures are a bit difficult to read. Additionally (and more importantly), do the data in these figures refer to the prefiltered data, or the final data used in the analyses? It certainly would help if you could provide both, to be able to assess the degree of filtering by SaTScan.

Line 106-107. “These outbreaks ... commercial farms”. It would be helpful if the authors could provide a definition for each premise type (villages, backyard farms, commercial farms) in the introduction, or mentioned elsewhere in the manuscript where relevant. As various countries/territories are included in the study, each perhaps using their own definition for backyard and villages farms, a consensus on these premise type definitions would strengthen the findings. In the analysis, backyard farms and villages were combined for HPAI, based on similar farming practices between the two, but not for ASF (line 449-451). What do the farming practices look like? If possible, it would be helpful to briefly explain why two premise types could be combined for HPAI by describing the farming practices for each premise type. Furthermore, are the routes for outbreak notification similar for backyard, villages and commercial farms? Could a barrier to report outbreaks for one of the premise types interfere with its surveillance performance? For instance, in case it is more difficult for a backyard farm to officially report an outbreak, delay in this process could result in lower odds of a zero-fatality rate.

Line 149. Typo: premise type -> Premise type

Line 185-189. “For HPAI outbreaks ... (Table 4)”. This sentence is unclear.

Line 192-194. “Finally, ... high-income countries/territories”. In this comparison outbreaks part of clusters in lower-middle income countries/territories are compared to outbreaks that were not part of clusters in high-income countries/territories. However, this does not seem to be a valid comparison for cluster vs. non-cluster outbreaks or lower-middle-income vs. high-income countries, as the groups differ in both aspects.

Line 213-215. “In contrast, ... (red bars in Fig. 2)”. Suggestion to remove ‘and’ (line 213), and add ‘fatalities’ after ‘more’ (line 214).

Line 225-227. “The countries/territories ... (Fig. S5)”. It is unclear from the following sentence whether countries that performed better tended to have a high zero-fatality probability and a low fatality slope because they performed better, or if they are classified as ‘performing better’ because they showed these trends. The same question applies to countries that performed worse and showed opposite trends for the fatality metrics.

Line 249-251. This is not obvious, at least to us. In fact, as long as local depletion of susceptible animals plays a minor role (and assuming frequency-dependent transmission), the number of infected animals is also only to a minor extent affected by premise size. Please clarify or leave out.

Line 236-345. “The zero-fatality probability ... and deaths”. Suggestion to move the description of the three fatality metrics to the introduction section. Additionally, further clarification in what way the fatality slope reflects the effectiveness of biosecurity measures and surveillance in detecting outbreaks would be appreciated (lines 237–238). Furthermore, the

fatality threshold is described differently in lines 240-241 from its initial description in lines 89-90.

Line 400-403. "In conclusion, ... improve these capacities". The authors highlight that they found a variability in animal health surveillance capacities within and between countries/territories, suggesting there is a need for efforts to improve these capacities. If possible, it would be interesting to highlight some suggestions of the authors on how to improve the surveillance capacities based on their outcomes (certain strategies that are currently only applied in countries with a high surveillance performance level).

Line 430-431. "The outbreaks ... country/territory)". In the text the authors refer to tables S3 and S4, which, in the supplementary information, are probably S1 and S2.

Line 442-443. "Clusters were ... ASF outbreaks". The reference (15) describes ASF outbreaks only. A space-time window of 180 days for HPAI seems excessive given the much faster dynamics of HPAI compared to ASF. In fact, for HPAI we found papers that use a time-window of 30 days (e.g., Ahmed SS et al (2010). The space-time clustering of highly pathogenic avian influenza (HPAI) H5N1 outbreaks in Bangladesh. *Epidemiology & Infection*, 138(6), 843-852), or 3 months (e.g., Marimwe M C et al (2021). The spatiotemporal epidemiology of high pathogenicity avian influenza outbreaks in key ostrich producing areas of South Africa. *Preventive Veterinary Medicine*, 196, 105474). Please clarify and analyze alternatives.

Line 452-454. But this can introduce bias. If it is deemed to be a minor factor (which may well be true), we believe some underpinning numbers are needed.

Line 490-491. Unclear. Please clarify.

Line 501. Effective number of samples of 500 is rather low, especially if the goal is to estimate 2.5% or 5% posterior tail probabilities. Why base model comparisons on DIC (which assumes a normal approximation of the posterior distribution), rather than the more general and more robust WIAC/LOO-IC?

Line 502-505. Why forward selection only? Please give rationale. What is the likelihood that in this manner the optimal model is not the one ultimately selected?

Line 535. The provided github page does not refer to the results in this manuscript. I believe this should be <https://github.com/KimYounjung/WOAHOutbreakSurveillance>. Stan and R code are well-structured, albeit code provided is minimal.

S6 and S7. Not all country IDs are in both figures. In addition, why is country ID 39 not indicated as 'performing worse' for backyard farms and villages?

Table 3. "Space-time outbreak cluster by income status". What was used as the reference group for 'In space-time outbreak cluster'? (b) seasons were defined in the opposite way for countries in the Southern Hemisphere. However, were considerations given to countries that do not experience four seasons but rather have rainy and dry seasons?

Recommendation

Based on the above evaluation, we suggest improving the current manuscript by providing more depth and clarification of the results. It would add value to explain how countries/territories could adjust their surveillance strategies to improve their performance. Once the issues addressed in this review are resolved, this manuscript could become a valuable and insightful contribution to the journal. However, we defer to the editors to determine if this is acceptable for Nature Communications.

(Remarks on code availability)

The link to the code is incorrect, but there is a repository that contains clean code. The code is minimal, it does not contain code to reproduce all results.

Reviewer #2

(Remarks to the Author)

(Remarks on code availability)

Reviewer #3

(Remarks to the Author)

The manuscript presents a study on the performance of surveillance systems for high pathogenicity avian influenza (HPAI) and African swine fever (ASF) outbreaks, utilizing data from the World Organisation for Animal Health (WOAH). The authors employ a novel modeling framework to assess outbreak notification performance based on reported fatalities. While the study addresses a critical area in animal health surveillance, several aspects require further attention to enhance the manuscript's clarity, rigor, and overall contribution to the field.

1. The study identifies significant variability in surveillance capacities for HPAI and ASF across different countries and territories. The focus on fatalities as a key performance indicator is a valuable contribution, as it underscores the importance of timely and accurate reporting in outbreak management. However, the manuscript would benefit from a more explicit discussion of the implications of these findings for policy and practice in animal health surveillance.
2. The work is relevant and timely, given the ongoing challenges posed by HPAI and ASF globally. It provides a framework that could inform improvements in surveillance systems, which is crucial for early detection and response. However, the authors should more clearly articulate how their findings compare to existing literature and what new insights they offer. A more thorough literature review would strengthen the manuscript's context.
3. While the study presents original findings, it is essential to ensure that the claims made are well-supported by the data. The authors should provide additional evidence or case studies that illustrate the practical implications of their modeling framework. This would enhance the credibility of their conclusions and demonstrate the framework's applicability in real-world scenarios.
4. The data analysis appears methodologically sound, but there are areas where the interpretation could be improved. For instance, the authors should address potential biases in the data collection process and discuss how these may impact the results. Additionally, a more detailed explanation of the statistical methods used in the modeling framework would enhance transparency and allow for better evaluation of the findings.
5. The methodology is generally appropriate for the research questions posed. However, the authors should provide more detail on the modeling framework, including the assumptions made and the rationale behind the selection of specific variables. This would facilitate reproducibility and allow other researchers to build upon this work.
6. While the methods are described, there is a need for greater detail to ensure that the study can be replicated. Specific information regarding data sources, analytical techniques, and the criteria for outbreak selection should be included. This would improve the manuscript's rigor and enhance its value to the research community.
7. The inclusion of reporting summaries would significantly improve the transparency and reproducibility of the results. Summaries that outline key findings, methodologies, and implications would provide readers with a clear understanding of the study's contributions.

(Remarks on code availability)

The provided R code implements Bayesian modeling using RStan for outbreak analysis, incorporating data on susceptible animals, deaths, and covariates such as country, season, epidemiological unit, and cluster membership. It preprocesses categorical variables using `model.matrix`, constructs an input list for the Stan model, runs the model with MCMC sampling, and saves the results.

While functional, the code has several areas for improvement in reproducibility and community usability. Specifically:

1. The code lacks an accurate README file to guide users through setup and execution. Missing details about dependencies, including R and RStan versions, as well as required R packages like `gridExtra`. The expected structure and contents of `data_sample.csv` are not specified, making it unclear how to format the input data. No explanation of the `.stan` model file (`WOAH_WAHIS.stan`) is provided, leaving users uncertain about the model's structure or objectives.
2. There are no checks for missing or invalid data in `data_sample.csv`. No scaling or transformation is included for variables that might require it (e.g., if the Stan model assumes certain distributions or scales).
3. While parallelization is configured, there are no diagnostic steps post-MCMC to assess model convergence, such as examining trace plots, R_{hat} values, or effective sample sizes. It assumes the `.stan` file is correctly defined but does not validate its compatibility with the provided data.
4. Without clear documentation and data specifications, reproducing results is challenging for external users. Missing context on how the code contributes to the research (e.g., specific hypotheses or outbreak characteristics being studied).

My advice is therefore (i) to include a README file detailing installation steps, dependencies, and data formatting requirements; (ii) to add preprocessing steps to validate and clean the input data; (iii) to provide information on the Stan model and expected results; (iv) to incorporate diagnostics and visualization of results to evaluate and interpret the adapted model; (v) to share example data sets with model output to facilitate understanding and replication.

Version 1:

Reviewer comments:

Reviewer #1

(Remarks to the Author)

Review of the second draft of manuscript 'Surveillance Performance in HPAI and ASF Outbreak Notification: Modelling Insights from WOAHS WAHIS Notification Data' by Younjung Kim and colleagues, submitted to Nature Communications.

We have read the revised manuscript and evaluated the responses by the authors to our earlier comments. In general we feel that the authors have addressed most comments adequately, and where they did not heed our suggestions they have explained why they did not follow our suggestions.

Overall, we are in particular pleased with the

- (1) restructuring of the results into distinct HPAI/ASF subsections;
- (2) explanation on how fatality metrics can guide improvements in surveillance;
- (3) clarification of the SaTScan usage and outbreak classification logic (even though we would have liked to see some more details on how the results are specifically affected by using short windows); and
- (4) the highlighting of data gaps (species-level detail, premise-type definitions).

In all, we are happy with the generally thorough responses and revisions of the manuscript.

(Remarks on code availability)

Documentation of the code is not extensive. code is well-structured and it should enable researchers redo these analyses (after downloading the relevant data).

Reviewer #2

(Remarks to the Author)

(Remarks on code availability)

Reviewer #3

(Remarks to the Author)

The revised manuscript has addressed many of the reviewers' concerns, particularly in clarifying the methodology, improving the presentation of results, and enhancing the reproducibility of the code; however, some aspects could still be strengthened, such as providing more concrete suggestions for how countries with lower surveillance performance might improve their systems, for instance by adapting successful practices from high-performing countries, and including a brief discussion on the lack of data regarding bird species in HPAI outbreaks and how this limitation may affect result interpretation and be mitigated in future studies; further, a more detailed discussion on how temporal changes in surveillance capacity over the study period might have influenced the findings and whether any adjustments were made would be helpful, as would an expanded reflection on how data standardization efforts could be implemented across countries and what impact this might have on evaluating surveillance performance; although the manuscript addresses the choice of temporal windows for cluster detection, a sensitivity analysis exploring how different window sizes affect the results would strengthen the robustness of the conclusions, and finally, the case of country or territory ID 37 (ASF outbreaks in villages) warrants deeper analysis to understand why this outlier occurred and what implications it may have for evaluating surveillance systems more broadly.

(Remarks on code availability)

Point-by-point responses to the reviewers' comments

The line numbers align with the manuscript with track changes. Please use the source file (.docx), not the PDF file (.pdf), to see track changes.

Reviewer #1 (Remarks to the Author):

General comments/evaluation

Comment 1.1: The topic of the study is highly relevant because of the increasing spillover threat of zoonotic HPAI H5N1. Many papers highlight the importance of infectious disease surveillance system strengthening, as part of 'pandemic preparedness' strategies. This manuscript aims to add to this goal using a modelling approach that assesses the performance of surveillance systems in countries reporting to the WOA. To this end, it uses the fatalities reported at initial outbreak notification as a performance measure. It was certainly an interesting and insightful read. However, we believe a more solution-oriented results section could have contributed to the value of this manuscript. The authors outline the factors (mainly premise size and country/territory) that play an important role in the performance of a surveillance system. However, no clear distinction is made between the surveillance strategies of countries that perform notably better or worse than average (Figures 2 and 3). If available, practical suggestions for improving surveillance in regions that score lower could add significant value to the manuscript.

The manuscript focuses on both the surveillance performance in HPAI as well as ASF outbreak performance. Both diseases have a significant impact on their susceptible hosts as well as on the economy. However, HPAI and ASF typically do not share host species, therefore, combining them does not provide additional insights in assessing the performance of the surveillance systems for each infectious disease separately. It does, however, sometimes make the text a bit more difficult to follow, or it sometimes results in the use of references which are only applicable to one of the two diseases. To improve the flow of the manuscript, or to prevent confusion, the work on both HPAI and ASF could be for instance be described in separate sub-chapters. This applies in particular to the Results section, which we believe could profit from main structuring by pathogen.

Response: We appreciate the reviewer's recognition of our manuscript's contribution to assessing the performance of animal infectious disease surveillance, as well as the reviewer's invaluable comments and suggestions, which have helped improve our manuscript through this revision.

In this comment, the reviewer makes the following two suggestions.

The first suggestion is to provide practical recommendations for improving surveillance in specific regions, which, in our case, correspond to countries/territories. While we fully acknowledge the importance of developing surveillance strategies tailored to individual countries/territories, the strength of this study indeed lies in its ability to identify which of them appear to be more or less effective, which has been challenged by the paucity of standardised

data. Therefore, our approach serves as an initial but foundational step in investigating the eco-social, veterinary, and agricultural factors driving the observed differences in surveillance performance between countries/territories. Such an investigation would require a targeted and systematic survey of these factors across the countries/territories being compared, which will be the natural next step following this study. Additionally, such a survey could also be used to gather more specific country/territory-level, as well as premise-level variables, to further refine our modelling results. We have now highlighted this throughout the revised manuscript.

Regarding the second suggestion to describe HPAI and ASF separately, we fully agree with the reviewer and have revised the Results section to present the findings more distinctly by pathogen. We have limited the revisions of the Discussion section in terms of presentation styles, as it was in the original manuscript as it facilitates a more general interpretation of the results.

Comment 1.2: Additionally, we believe that the steps used to define clusters using SaTScan analysis could and should have been explained better, and also should be presented in the Results section. How do the results depend on this analysis? In particular, is a time window of 180 days for both ASF and HPAI plausible given the differences in the outbreak dynamics of these pathogens? Are alternative maximum radiuses considered? Can you stratify the data by cluster versus non-cluster (e.g., in Figures S1 and S2)?

Response: SatScan was used to classify whether a given outbreak occurred simultaneously in time and space along other outbreaks in a given country/territory. We set 180 days as the maximum temporal window size to allow for the detection of clusters occurring within this period. However, this does imply that all clusters span the full 180 days.

Our choice of this window size was based on the variability in outbreak cluster pressures on veterinary authorities across different countries and territories, as noted by the reviewer. These pressures depend on factors such as surveillance and control capacities and the nature of different pathogens. However, a shorter window (e.g., 30 days or 3 months) might fail to capture outbreak clusters that persist beyond this period, even though they continue to exert pressure on veterinary services and impact surveillance performance. That is, both long, protracted clusters and short, intense clusters would place pressure on veterinary authorities.

Considering this, by allowing a broader window while still limiting it to half a year to account for disease seasonality, we aimed to capture such variations and ensure that all relevant clusters were detected. In fact, the majority of the identified clusters spanned between 1 and 3.5 months (interquartile range), with a median of 2 months. This suggests that most of these clusters ranged a shorter window the reviewer considered more suitable. We have now included more detailed SatScan results, though not extensively, as our focus remains on the main modelling framework results.

Lines 120 – 123

Figures S1 and S2

We would also like to note that during this revision, we identified an error in the cluster classification of outbreaks. The revised manuscript presents results based on the corrected data; however, this did not affect the overall interpretation of the study findings. Finally, we would also like to clarify here that, in response to the reviewer's comment below (Comment 1.9), that we did not use SatScan to filter (i.e. remove) any outbreaks from the analysis. Rather, each outbreak was simply classified as either occurring within an outbreak cluster or not.

Comment 1.3: On another note, it is important to take the susceptibility of different bird species for HPAI into account when scoring the surveillance performance on fatality counts. While the authors do mention the potential effect of host breeds and species (lines 341-346) on the variability in virus susceptibility – and thus, performance of outbreak notification within the same country/territory – they did not include the difference in host susceptibility in their model. Where both ducks and poultry are included in 'birds', HPAI appears to be more fatal to poultry than ducks. This could perhaps lead to lower zero-fatality odds for notifications from poultry farms compared to duck farms. In countries/territories where more poultry than ducks are kept, this factor might influence the overall performance level of the surveillance applied.

Response: We fully agree with the reviewer that host species can affect the surveillance performance, especially for HPAI. We discussed the importance of this aspect when designing the present study. However, unfortunately, bird and swine species information were not available in the WOAHS WAHIS. We have revised the manuscript to further discuss its potential impacts on the study results.

Lines 123 – 124, 479 – 493

Comment 1.4: Overall, we found no errors in the logic or statistical analyses in this study and believe the results hold significant public health implications. However, the anonymization of countries (identified only by numbers) prevents detailed analyses and specific recommendations for individual countries. We defer to the editors to determine if this is acceptable for Nature Communications.

Response: We conducted this study in collaboration with the WOAHS, with two of the co-authors affiliated with the organisation. The decision to anonymise the country names was made in accordance with WOAHS guidance.

Lines 119 – 120, 644

Specific comments

Comment 1.5: Line 37. The reference (1) is used to describe both HPAI and ASF, but it only discusses ASF. It might be helpful to find a similar reference specifically for HPAI.

Response: The following reference for HPAI is now added to the revised manuscript:

Beato, M. S. & Capua, I. Transboundary spread of highly pathogenic avian influenza through poultry commodities and wild birds: a review. Rev Sci Tech 30, 51-61 (2011).

<https://doi.org/10.20506/rst.30.1.2013>

Line 37

Comment 1.6: Line 76-78. “Secondly, ... of the exposed population”. It is stated that larger susceptible host populations are likely to experience more fatalities due to the size of the exposed population. Are fatalities being measured in absolute numbers or proportions? For instance, while the absolute number of fatalities may be higher in larger populations than smaller populations, the proportion of fatalities is likely to be lower compared to smaller populations.

Response: We measured fatalities in absolute numbers. We agree with the reviewer that the proportion of fatalities can display different patterns from the absolute number of fatalities. In our modelling framework, the fatality slope metric is defined as the rate at which the number of fatalities at notification increases with premise size before plateauing. Therefore, it captures the relationship between the number of fatalities and the number of total animals. In this revision, we have summarised the definition, assumptions, and implications of each fatality metric in Table 1, rather than in text form, to enhance understanding of the modelling framework alongside Figure 1.

Comment 1.7: Line 99-101. “Finally, ... relative to others.” It would be helpful if the authors could clarify what is meant by ‘others.’ Were the comparisons made between different countries/territories, or between various groups?

Response: We have revised the sentence to ensure its meaning is accurately conveyed.

Lines 108 – 111

Comment 1.8: Line 103-109. We believe a more complete description of the data is in order, together with the results of the SaTScan analysis.

Response: The data sources and cleaning processes are detailed in the Methods section. We have included additional SaTScan results in the Results section and have revised Figures S1 and S2 to better illustrate temporal outbreak patterns based on spatiotemporal clustering. However, the revised manuscript remains focused on the results of the modelling framework, as the SaTScan results are not the primary study findings but rather serve as one of the independent variables.

Lines 120 – 123

Lines 564 – 580

Comment 1.9: Line 103-106. “The analysis ... (Figs. S1 and S2)”. The x-axes of the figures

are a bit difficult to read. Additionally (and more importantly), do the data in these figures refer to the prefiltered data, or the final data used in the analyses? It certainly would help if you could provide both, to be able to assess the degree of filtering by SaTScan.

Response: The figures are updated as requested. These figures show the distribution of the final data. Showing the pre-processed data contain multiple reports from the same outbreaks due to the way Member countries/territories make outbreak notifications to the WOA. Therefore, its distribution would not provide meaningful information. Additionally, we did not use SaTScan to filter outbreak data but to determine whether a given outbreak was included in a spatiotemporal outbreak cluster or not.

Figures S1 and S2

Comments 1.10: Line 106-107. “These outbreaks ... commercial farms”. It would be helpful if the authors could provide a definition for each premise type (villages, backyard farms, commercial farms) in the introduction, or mentioned elsewhere in the manuscript where relevant. As various countries/territories are included in the study, each perhaps using their own definition for backyard and villages farms, a consensus on these premise type definitions would strengthen the findings.

Response: We searched both the WOA notification procedure and the Terrestrial Code Glossary but could not find specific definitions for these premise types. We fully agree with the reviewer that establishing a consensus on premise types would enhance the robustness of our findings. In the revised manuscript, we have now discussed this as an example highlighting the need for more standardised reporting practices.

Lines 508 – 511

Comment 1.11: In the analysis, backyard farms and villages were combined for HPAI, based on similar farming practices between the two, but not for ASF (line 449-451). What do the farming practices look like? If possible, it would be helpful to briefly explain why two premise types could be combined for HPAI by describing the farming practices for each premise type.

Response: The primary reason for combining these two premise types was that there were only two countries/territories that reported HPAI outbreaks in villages. Then, considering the similarity of villages with backyard farms (e.g. small scale, birds are likely to be allowed to roam freely, and limited veterinary care), rather than with commercial farms, we have decided to combine outbreaks from villages and backyard farms.

The manuscript has been revised to clarify this.

Lines 583 – 587

Comment 1.12: Furthermore, are the routes for outbreak notification similar for backyard, villages and commercial farms? Could a barrier to report outbreaks for one of the premise types

interfere with its surveillance performance? For instance, in case it is more difficult for a backyard farm to officially report an outbreak, delay in this process could result in lower odds of a zero-fatality rate.

Response: This would vary depending on the surveillance systems of each country/territory, and our results likely reflect these potential differences. As a next step, for a given country/territory, our findings could help facilitate discussions on the factors influencing surveillance performance in outbreak notification compared to other countries/territories. Information on the routes of outbreak notification could be gathered through this process.

Comment 1.13: Line 149. Typo: premise type -> Premise type

Response: The manuscript has been revised accordingly.

Comment 1.14: Line 185-189. “For HPAI outbreaks ... (Table 4)”. This sentence is unclear.

Response: The manuscript has been revised accordingly.

Lines 227 – 230

Comment 1.15: Line 192-194. “Finally, ... high-income countries/territories”. In this comparison outbreaks part of clusters in lower-middle income countries/territories are compared to outbreaks that were not part of clusters in high-income countries/territories. However, this does not seem to be a valid comparison for cluster vs. non-cluster outbreaks or lower-middle-income vs. high-income countries, as the groups differ in both aspects.

Response: We agree with the reviewer and have revised the manuscript accordingly.

Lines 255 – 259

Comment 1.16: Line 213-215. “In contrast, ... (red bars in Fig. 2)”. Suggestion to remove ‘and’ (line 213), and add ‘fatalities’ after ‘more’ (line 214).

Response: The manuscript has been revised accordingly.

Line 291 – 292

Comment 1.17: Line 225-227. “The countries/territories ... (Fig. S5)”. It is unclear from the following sentence whether countries that performed better tended to have a high zero-fatality probability and a low fatality slope because they performed better, or if they are classified as ‘performing better’ because they showed these trends. The same question applies to countries that performed worse and showed opposite trends for the fatality metrics.

Response: Countries/territories were classified as performing significantly better or worse by comparing their observed fatalities with the expected fatalities, assuming the average performance across all countries/territories. After identifying these countries/territories, we examined their fatality metric estimates to understand the factors influencing their performance relative to others. The manuscript has been revised to clarify this process.

Lines 271 – 275

Comment 1.18: Line 249-251. This is not obvious, at least to us. In fact, as long as local depletion of susceptible animals plays a minor role (and assuming frequency-dependent transmission), the number of infected animals is also only to a minor extent affected by premise size. Please clarify or leave out.

Response: Our modelling framework is based on the assumption that HPAI and ASF transmission is primarily density-dependent, a common consideration for farm animal disease transmission. Upon pathogen introduction, transmission begins within groups of animals housed together. Larger premises are likely to accommodate more animals per group within limited space, leading to higher contact rates and, consequently, faster transmission. This assumption is now explained in Table 1.

Comment 1.19: Line 236-345. “The zero-fatality probability ... and deaths”. Suggestion to move the description of the three fatality metrics to the introduction section. Additionally, further clarification in what way the fatality slope reflects the effectiveness of biosecurity measures and surveillance in detecting outbreaks would be appreciated (lines 237–238). Furthermore, the fatality threshold is described differently in lines 240-241 from its initial description in lines 89-90.

Response: We have now revised and relocated the description of the three fatality metrics to the introduction section, as the reviewer suggested. Notably, we have now inserted Table 1 to better describe the definition, assumptions, and implications of each fatality metric along with Figure 1.

Table 1 and Figure 1

Comment 1.20: Line 400-403. “In conclusion, ... improve these capacities”. The authors highlight that they found a variability in animal health surveillance capacities within and between countries/territories, suggesting there is a need for efforts to improve these capacities. If possible, it would be interesting to highlight some suggestions of the authors on how to improve the surveillance capacities based on their outcomes (certain strategies that are currently only applied in countries with a high surveillance performance level).

Response: A high level of heterogeneities in animal health surveillance capacities suggests the influence of various factors that were not fully captured by the macro-level variables used in

this study. Specific recommendations for improving animal health surveillance capacities could only be made after identifying these unaccounted-for factors.

By highlighting significant disparities in surveillance capacities, our findings provide a foundation for this next step. For example, global or regional efforts could focus on understanding key differences in the eco-social contexts of countries/territories that have performed exceptionally well or poorly compared to others. Identifying these differences will help uncover their underlying drivers, ultimately enabling the development of targeted, effective, and efficient recommendations.

The manuscript has been revised to clarify our arguments.

Lines 335 – 339, 521 – 529

Comment 1.21: Line 430-431. “The outbreaks ... country/territory)”. In the text the authors refer to tables S3 and S4, which, in the supplementary information, are probably S1 and S2.

Response: We appreciate the reviewer for pointing out this mistake. The manuscript has been revised accordingly to ensure the correct reference to the supplementary tables.

Line 557 – 558

Comment 1.22: Line 442-443. “Clusters were ... ASF outbreaks”. The reference (15) describes ASF outbreaks only. A space-time window of 180 days for HPAI seems excessive given the much faster dynamics of HPAI compared to ASF. In fact, for HPAI we found papers that use a time-window of 30 days (e.g., Ahmed SS et al (2010). The space–time clustering of highly pathogenic avian influenza (HPAI) H5N1 outbreaks in Bangladesh. *Epidemiology & Infection*, 138(6), 843-852), or 3 months (e.g., Marimwe M C et al (2021). The spatiotemporal epidemiology of high pathogenicity avian influenza outbreaks in key ostrich producing areas of South Africa. *Preventive Veterinary Medicine*, 196, 105474). Please clarify and analyze alternatives.

Response: We set 180 days as the maximum temporal window size, meaning it allows for the detection of clusters occurring within this period but does not necessarily assume all clusters span the full 180 days.

Our choice was based on the variability in outbreak cluster pressures on veterinary authorities across different countries/territories, depending on their surveillance and control capacities. A shorter window (e.g., 30 days or 3 months) might fail to capture outbreak clusters that persist beyond this period, even though they continue to exert pressure on veterinary services, thereby impacting surveillance performance. By allowing a broader window, we aimed to account for such variations and ensure that all relevant clusters were detected.

Additionally, we believe that using a shorter window would have minimal impact on the study results, as most identified clusters spanned between 1 and 3.5 months (interquartile range), with a median duration of 2 months. This range indeed covers shorter temporal windows the

above papers the reviewer used as examples. We have now included this information to help readers assess the potential impact of temporal cluster window size. Also, a new reference regarding HPAI has now been added.

Lines 564 – 580

Comment 2.23: Line 452-454. But this can introduce bias. If it is deemed to be a minor factor (which may well be true), we believe some underpinning numbers are needed.

Response: We have now provided the information on the distribution of outbreaks per country/territory.

Lines 118 – 120

Comment 2.24: Line 490-491. Unclear. Please clarify.

Response: The sentence has now been clarified.

Lines 634 – 636

Comment 2.25: Line 501. Effective number of samples of 500 is rather low, especially if the goal is to estimate 2.5% or 5% posterior tail probabilities. Why base model comparisons on DIC (which assumes a normal approximation of the posterior distribution), rather than the more general and more robust WIAC/LOO-IC?

Response: Following the reviewer's suggestion, we have revised our model comparison based on LOOIC. For ASF outbreaks, the full model explained the fatality data significantly better than other models and was therefore selected as the best-fitting model. For HPAI outbreaks, no significant differences were detected between several models, including the full model. However, since several categories within the variables of the full model were significantly associated with the fatality metrics, the full model was selected as the best-fitting model. Selecting the full models as the best-fitting models for both HPAI and ASF outbreaks also facilitated the presentation of our findings.

Lines 143 – 144, 651 – 654

Tables 2 and 4

Regarding ESS, it is indeed well over 1000. The manuscript has been revised accordingly.

Line 648

Comment 2.26: Line 502-505. Why forward selection only? Please give rationale. What is the likelihood that in this manner the optimal model is not the one ultimately selected?

Response: A stepwise forward selection process was chosen because it allowed for tracking the increase in deviance R^2 as additional variables were added to the model. This approach enabled us to assess the incremental contribution of individual variables in explaining the data, as you can see in Tables 1 and 2.

Furthermore, the variables not selected in the best-fitting models were those associated with spatiotemporal cluster for both HPAI and ASF outbreaks: i) *Outbreak Cluster* and ii) *Outbreak Cluster by Veterinary Service Capacity*. Please note that due to their correlation, only one of these variables —along with iii) *Outbreak Cluster by Country/Territory Income Level*—could be considered in the same model. Among these, iii) *Outbreak Cluster by Country/Territory Income Level* outperformed the other two, while its contribution was smaller than other variables, such as *country* and *premise type*. Therefore, we believe that alternative model selection processes would have resulted in the same variables being included in the best-fitting models.

Comment 2.27: Line 535. The provided github page does not refer to the results in this manuscript. I believe this should be <https://github.com/KimYounjung/WOAHOutbreakSurveillance>. Stan and R code are well-structured, albeit code provided is minimal.

Response: It appears that a non-updated version of the manuscript has been provided for peer-review. The revised manuscript contains the correct link.

Comment 2.28: S6 and S7. Not all country IDs are in both figures. In addition, why is country ID 39 not indicated as ‘performing worse’ for backyard farms and villages?

Response: Not all countries/territories reported outbreaks for all premise types, which is why not all country/territory IDs are shown in both figures. Countries/territories were classified as performing significantly better or worse if the 95th percentile intervals [PI] predicted by the best-fitting model did not overlap with the 95th PI predicted by the model excluding country/territory. Consequently, country ID 39 was not classified as performing significantly worse. This criterion is explained in the legends of Figures 2-3 and Figures S3-S7.

Comment 2.29: Table 3. “Space-time outbreak cluster by income status”. What was used as the reference group for ‘In space-time outbreak cluster’? (b) seasons were defined in the opposite way for countries in the Southern Hemisphere. However, were considerations given to countries that do not experience four seasons but rather have rainy and dry seasons?

Response: Outbreaks that were not part of clusters in high-income countries/territories served as the reference group, even for outbreaks within clusters. This approach was used because cluster membership and income status were combined to define a category. Regarding season, the countries/territories included from the Southern Hemisphere were those with relatively distinct four-season patterns. Therefore, we believe this classification had a limited impact on the model results

Recommendation

Comment 2.30: Based on the above evaluation, we suggest improving the current manuscript by providing more depth and clarification of the results. It would add value to explain how countries/territories could adjust their surveillance strategies to improve their performance. Once the issues addressed in this review are resolved, this manuscript could become a valuable and insightful contribution to the journal. However, we defer to the editors to determine if this is acceptable for Nature Communications.

Response: We truly appreciate the reviewer's constructive and detailed comments and suggestions. We believe the manuscript has improved significantly through this review.

Reviewer #1 (Remarks on code availability):

Comment 2.31: The link to the code is incorrect, but there is a repository that contains clean code. The code is minimal, it does not contain code to reproduce all results.

Response: The revised manuscript includes the correct link. Additionally, this link provides expanded model codes with more detailed information on sample data, model execution, and validation.

Reviewer #2 (Remarks to the Author):

Comment 2.32: I co-reviewed this manuscript with one of the reviewers who provided the listed reports. This is part of the Nature Communications initiative to facilitate training in peer review and to provide appropriate recognition for Early Career Researchers who co-review manuscripts.

Response: We truly appreciate the reviewer's constructive and detailed comments and suggestions. We believe the manuscript has improved significantly through this review.

Reviewer #3 (Remarks to the Author):

Comment 3.1: The manuscript presents a study on the performance of surveillance systems for high pathogenicity avian influenza (HPAI) and African swine fever (ASF) outbreaks, utilizing data from the World Organisation for Animal Health (WOAH). The authors employ a novel modeling framework to assess outbreak notification performance based on reported fatalities. While the study addresses a critical area in animal health surveillance, several aspects require further attention to enhance the manuscript's clarity, rigor, and overall contribution to the field.

1. The study identifies significant variability in surveillance capacities for HPAI and ASF across different countries and territories. The focus on fatalities as a key performance indicator is a valuable contribution, as it underscores the importance of timely and accurate reporting in

outbreak management. However, the manuscript would benefit from a more explicit discussion of the implications of these findings for policy and practice in animal health surveillance.

2. The work is relevant and timely, given the ongoing challenges posed by HPAI and ASF globally. It provides a framework that could inform improvements in surveillance systems, which is crucial for early detection and response. However, the authors should more clearly articulate how their findings compare to existing literature and what new insights they offer. A more thorough literature review would strengthen the manuscript's context.

3. While the study presents original findings, it is essential to ensure that the claims made are well-supported by the data. The authors should provide additional evidence or case studies that illustrate the practical implications of their modeling framework. This would enhance the credibility of their conclusions and demonstrate the framework's applicability in real-world scenarios.

4. The data analysis appears methodologically sound, but there are areas where the interpretation could be improved. For instance, the authors should address potential biases in the data collection process and discuss how these may impact the results. Additionally, a more detailed explanation of the statistical methods used in the modeling framework would enhance transparency and allow for better evaluation of the findings.

5. The methodology is generally appropriate for the research questions posed. However, the authors should provide more detail on the modeling framework, including the assumptions made and the rationale behind the selection of specific variables. This would facilitate reproducibility and allow other researchers to build upon this work.

6. While the methods are described, there is a need for greater detail to ensure that the study can be replicated. Specific information regarding data sources, analytical techniques, and the criteria for outbreak selection should be included. This would improve the manuscript's rigor and enhance its value to the research community.

7. The inclusion of reporting summaries would significantly improve the transparency and reproducibility of the results. Summaries that outline key findings, methodologies, and implications would provide readers with a clear understanding of the study's contributions.

Response: First of all, we sincerely appreciate the reviewer's detailed and constructive feedback. We believe that addressing these comments has significantly strengthened the manuscript. Below, we outline the specific revisions made in response to each point.

Regarding the implications for policy and practice, we recognize the importance of discussing how our findings inform policy and practice in animal health surveillance. Our study highlights variability in surveillance performance across countries/territories, providing a foundational step toward identifying eco-social, veterinary, and agricultural factors influencing this variability. Understanding these drivers will ultimately enable the design of context-specific surveillance strategies. We have now emphasized these aspects in the revised manuscript.

Lines 336 – 339, 521 – 529

We have also added the strengthen of our approach, highlighting the limitations of implementing systematic evaluation of surveillance systems, with several seminal literature on animal disease surveillance system evaluation.

Lines 314 – 323, 335 – 339

To improve clarity, we have now included more detailed information on the modelling framework. For example, we have summarised the definition, assumptions, and implications of each fatality metric in Table 1 to help understand the modelling framework along with Figure 1. Additionally, we have now included more detailed information on how the fatality metrics were derived mathematically in the Method section. Additionally, we have expanded the limitations of our study to discuss potential bias arising from our modelling framework and data used.

Lines 473 – 520, 590 - 680

Details on data sources and outbreak selection criteria are now explicitly stated in the Methods section. Regarding a reporting summary, while Nature Communications does not allow a separate reporting summary, we have restructured the introduction to our modelling framework, ensuring clarity through Table 1 and Figure 1.

Reviewer #3 (Remarks on code availability):

Comment 3.2: The provided R code implements Bayesian modeling using RStan for outbreak analysis, incorporating data on susceptible animals, deaths, and covariates such as country, season, epidemiological unit, and cluster membership. It preprocesses categorical variables using `model.matrix`, constructs an input list for the Stan model, runs the model with MCMC sampling, and saves the results.

While functional, the code has several areas for improvement in reproducibility and community usability. Specifically:

1. The code lacks an accurate README file to guide users through setup and execution. Missing details about dependencies, including R and RStan versions, as well as required R packages like `gridExtra`. The expected structure and contents of `data_sample.csv` are not specified, making it unclear how to format the input data. No explanation of the `.stan` model

file (WOAH_WAHIS.stan) is provided, leaving users uncertain about the model's structure or objectives.

2. There are no checks for missing or invalid data in data_sample.csv. No scaling or transformation is included for variables that might require it (e.g., if the Stan model assumes certain distributions or scales).

3. While parallelization is configured, there are no diagnostic steps post-MCMC to assess model convergence, such as examining trace plots, Rhat values, or effective sample sizes. It assumes the .stan file is correctly defined but does not validate its compatibility with the provided data.

4. Without clear documentation and data specifications, reproducing results is challenging for external users. Missing context on how the code contributes to the research (e.g., specific hypotheses or outbreak characteristics being studied).

My advice is therefore (i) to include a README file detailing installation steps, dependencies, and data formatting requirements; (ii) to add preprocessing steps to validate and clean the input data; (iii) to provide information on the Stan model and expected results; (iv) to incorporate diagnostics and visualization of results to evaluate and interpret the adapted model; (v) to share example data sets with model output to facilitate understanding and replication.

Response: We sincerely appreciate the reviewer's detailed feedback and constructive suggestions regarding code availability. We fully acknowledge the importance of well-documented, reproducible code for model understanding and replication. Accordingly, we have now updated the shared files on GitHub. Below, we address each point in detail:

Firstly, we apologise for an oversight in the preparation of files shared in the Github. The name of the RStan file has now been corrected.

Regarding preprocessing and validation, we have anonymised sample data for model fitting in compliance with WOAHS guidelines. Therefore, the provided code does not include explicit steps for checking missing or invalid data based on raw data as it could reveal country/territory information. However, as the reviewer noted, premise size was highly right-skewed and required log transformation before modelling. This transformation was implemented during model fitting (see "RStan_WOAHS_WAHIS.stan"), and we have now explicitly explained this step in the README file.

Regarding model diagnostics and convergence checks, we have incorporated key model diagnostic components into the 'Run_WOAHS_WAHIS.R' code, including:

1. Assessment of Rhat values, and effective sample sizes
2. Assessment of traceplots, with the expected plot shown in the README file.

3. A posterior predictive check (PPC), with the corresponding code contained in the same code and the expected plot shown in the README file.

The authors have fitted the RStan model to the sample data and confirmed that all diagnostic indicators suggest model convergence.

Lastly, improved documentation of model structure and expected results, the updated README file now includes:

1. A detailed description of each variable in the sample data
2. An explanation of expected model validation results

Additionally, we have revised the Method section of the main manuscript to provide greater detail on the model structure and its application to outbreak surveillance.

We appreciate the reviewer's insightful comments, which have significantly improved the clarity and reproducibility of our shared code.

Point-by-point responses to the reviewer' comments

The line numbers align with the manuscript with track changes.

Reviewer 3 comments

Comment 1: The revised manuscript has addressed many of the reviewers' concerns, particularly in clarifying the methodology, improving the presentation of results, and enhancing the reproducibility of the code;

Response: We appreciate that the reviewer recognises our efforts to address the reviewers' comments. We believe that the revision process has significantly improved the quality of our manuscript. We also appreciate the reviewer's additional comments and suggestions, which we have responded below point by point.

Comment 2: however, some aspects could still be strengthened, such as providing more concrete suggestions for how countries with lower surveillance performance might improve their systems, for instance by adapting successful practices from high-performing countries,

Response: The first step toward making recommendations for countries with lower surveillance performance may be to identify key differences in biosecurity, surveillance, and outbreak control practices, along with a range of other contributing factors. These countries could potentially enhance their surveillance performance by adopting effective practices observed in countries with stronger surveillance systems.

We acknowledge, however, that this effort is not without challenges, as elements of biosecurity, surveillance, and outbreak control are closely linked to each country's broader socioeconomic context. Nevertheless, identifying these differences remains a necessary first step toward improving surveillance performance standards.

We have expanded on this point and highlighted the complexity in the revised text as follows:

Lines 271 – 281: “For these countries/territories, rather than simply attempting to transplant practices deemed successful in one setting to another, gathering detailed information on the eco-social, veterinary, and agricultural factors and assessing it in relation to differences in fatality metrics could help identify the key drivers of surveillance performance. In this process, it is important to keep in mind that the fatality metrics capture only a limited dimension of animal health systems and therefore cannot be used as the sole measure of their overall performance. This approach would then help understand how different contexts influence the actual operation of surveillance, providing a critical foundation for developing context-specific strategies to strengthen animal infectious disease surveillance at national, regional, and global levels.”

Comment 3: and including a brief discussion on the lack of data regarding bird species in HPAI outbreaks and how this limitation may affect result interpretation and be mitigated in future studies;

Response: The lack of data regarding bird species and its potential impact on study results were raised by Reviewers 1 and 2 during the previous revision. We have clarified such data gap in the revised manuscript and also discussed in depth how differential susceptibility by bird species would impact fatality metrics as follows:

Lines 395 – 410: “Secondly, the host animals in HPAI outbreaks were categorised as birds in the WOAHA WAHIS. HPAI viruses, however, are known to pose varying levels of transmissibility and virulence to different bird species. Notably, chickens are generally more susceptible than ducks to HPAI viruses^{25,26}. This suggests that our fatality metrics could be influenced by the composition of bird species within a given country. In chicken farms, a sudden rise in mortality can serve as an early warning sign of an outbreak, but by the time it is detected, substantial losses may have already occurred, potentially leading to a low zero-fatality probability and a high fatality slope. In contrast, HPAI outbreaks in duck farms may spread widely before detection with a few or no deaths, potentially resulting in a high zero-fatality probability and a low fatality slope. Therefore, having information on host species could help improve the assessment of surveillance performance in HPAI outbreak notification, ensuring that differences in species susceptibility and transmissibility are accounted for. For ASF outbreaks, accounting for viral genotypes and swine breeds would also allow more robust parameter estimation, although these concerns may be less relevant compared to the HPAI host/viruses, given the relatively stable viral genetic structure and the less variable transmissibility and virulence across different swine breeds^{27,28}.”

In addition, we have included a new paragraph at the end of the discussion section that suggests directions for future studies, which include incorporating interviews with individual veterinary services to obtain more detailed epidemiological data, including host species, and to assess potential biases in the data, as follows:

Lines 440 – 451: “Considering the limitations discussed above, future research would benefit from incorporating interviews with individual veterinary services to better understand differences in surveillance and reporting practices that may introduce biases into the data. These interviews could also help obtain more detailed outbreak notification data—such as host species, viral lineages, and other key epidemiological variables, including temporal changes in surveillance capacity—collected in a way that allows for meaningful comparisons within and between countries or territories. Not only would such efforts enable more robust comparisons between countries/territories, but they could also facilitate the application of our framework at the sub-national level, helping to identify heterogeneities within countries/territories and develop more tailored, context-specific recommendations. Crucially, these activities should be conducted in close collaboration with the veterinary services involved, to ensure accurate interpretation of findings and to establish a constructive feedback loop that supports the continuous strengthening of surveillance and response systems”

Comment 4: further, a more detailed discussion on how temporal changes in surveillance capacity over the study period might have influenced the findings and whether any adjustments were made would be helpful, as would an expanded reflection on how data standardization efforts could be implemented across countries and what impact this might have on evaluating surveillance performance;

Response: Regarding data standardisation efforts, we have included an additional paragraph to suggest directions for future studies, with an emphasis on conducting interviews with individual veterinary services (please see the response to Comment 3, and Lines 437 – 448).

Regarding temporal changes in surveillance capacity, we have now included the following part:

Lines 411 – 420: “Thirdly, surveillance capacity could have changed dramatically during the study periods within countries/territories as the level of viral circulation varied, veterinary services gained experience with newly circulating viruses, and socioeconomic and geopolitical contexts evolved. Such changes may have occurred in either direction, making it difficult to incorporate into our framework given limited data availability. Although we used the number of veterinarians per a million terrestrial animal biomass²⁹ as a proxy for veterinary service capacity, this metric does not reflect potential temporal shifts in the reporting behaviour of key stakeholders, including veterinarians, farmers, and other value chain actors. Therefore, future studies should aim to complement quantitative data with detailed descriptions of the contextual factors, such as politics, socioeconomic, and veterinary system dynamics, that may have influenced these behaviours and, ultimately, surveillance performance³⁰.”

Comment 5: although the manuscript addresses the choice of temporal windows for cluster detection, a sensitivity analysis exploring how different window sizes affect the results would strengthen the robustness of the conclusions,

Response: We set 180 days as the maximum temporal window to allow for the detection of clusters occurring within this period. We emphasise that this does not imply that all clusters span the full 180 days. In fact, the majority of identified clusters lasted between 1 and 3.5 months (interquartile range), with a median of 2 months.

We would also like to underscore that our choice of the 180-day window was based on the fact that the level of outbreak cluster pressures experienced by veterinary authorities is likely to vary across different countries/territories, depending on their surveillance and control capacities. A shorter window (e.g., 30 days or 3 months) might fail to capture outbreak clusters that persist beyond this period (e.g. long, protracted clusters), even though they continue to exert pressure on veterinary services, thereby impacting surveillance performance. By allowing a broader window, we aimed to account for such variations and ensure that all relevant clusters were detected (e.g. short, intense clusters as well as long, protracted clusters).

We have now revised the manuscript to clarify the logic behind choosing the maximum temporal window as well as make suggestions for future studies in a separate paragraph as follows:

Lines 353 – 361: “It is important to note that outbreak clusters were identified based on both HPAI and ASF outbreaks, under the assumption that they exert equal pressure on veterinary authorities. However, the pressure exerted by HPAI and ASF outbreaks, as well as other pathogens, on veterinary authorities likely varies between countries/territories depending on multiple factors, including their surveillance and control capacities. We therefore used 180 days as the maximum temporal window to allow for the detection of all relevant clusters within the study period (e.g. short, intense clusters as well as long, protracted clusters), while still considering seasonal HPAI and ASF transmission patterns. Shorter maximum windows might fail to capture long, protracted clusters that place significant pressure on veterinary services”

Comment 6: and finally, the case of country or territory ID 37 (ASF outbreaks in villages) warrants deeper analysis to understand why this outlier occurred and what implications it may have for evaluating surveillance systems more broadly.

Response: We have revised the manuscript to further expand the investigation of the outlier behaviour of country/territory 37, making step by step suggestions.

Lines 293 – 299: “Such investigations could therefore begin by assessing potential systematic differences in reporting practices, for example, how village outbreaks were defined and reported. Although outbreaks were notified using a standardised format, there is no consensus on the definition of the premise types, which may lead to inconsistencies in classification, particularly between outbreaks in backyard farms and villages. If no significant differences are identified, the investigation could then focus on the epidemiological context of these villages and how surveillance systems operated there, compared to those in other countries/territories.”